# ADAPTIVE $Q$-NETWORK: ON-THE-FLY TARGET SELECTION FOR DEEP REINFORCEMENT LEARNING

**Théo Vincent**[1,2,*]   **Fabian Wahren**[1,2]   **Jan Peters**[1,2,3]
**Boris Belousov**[1]   **Carlo D'Eramo**[2,3,4]

[1]DFKI GmbH, SAIROL   [2] Department of Computer Science, TU Darmstadt
[3] Hessian.ai, TU Darmstadt   [4]Center for AI and Data Science, University of Würzburg

## ABSTRACT

Deep Reinforcement Learning (RL) is well known for being highly sensitive to hyperparameters, requiring practitioners substantial efforts to optimize them for the problem at hand. This also limits the applicability of RL in real-world scenarios. In recent years, the field of automated Reinforcement Learning (AutoRL) has grown in popularity by trying to address this issue. However, these approaches typically hinge on additional samples to select well-performing hyperparameters, hindering sample-efficiency and practicality. Furthermore, most AutoRL methods are heavily based on already existing AutoML methods, which were originally developed neglecting the additional challenges inherent to RL due to its non-stationarities. In this work, we propose a new approach for AutoRL, called Adaptive $Q$-Network (AdaQN), that is tailored to RL to take into account the non-stationarity of the optimization procedure without requiring additional samples. AdaQN learns several $Q$-functions, each one trained with different hyperparameters, which are updated online using the $Q$-function with the smallest approximation error as a shared target. Our selection scheme simultaneously handles different hyperparameters while coping with the non-stationarity induced by the RL optimization procedure and being orthogonal to any critic-based RL algorithm. We demonstrate that AdaQN is theoretically sound and empirically validate it in MuJoCo control problems and Atari 2600 games, showing benefits in sample-efficiency, overall performance, robustness to stochasticity and training stability. Our code is available at *https://github.com/theovincent/AdaDQN*.

## 1 INTRODUCTION

Deep Reinforcement Learning (RL) has proven effective at solving complex sequential decision problems in various domains (Mnih et al., 2015; Haarnoja et al., 2018; Silver et al., 2017). Despite their success in many fields, deep RL algorithms suffer from brittle behavior with respect to hyperparameters (Mahmood et al., 2018; Henderson et al., 2018). For this reason, the field of automated Reinforcement Learning (AutoRL) (Parker-Holder et al., 2022) has gained popularity in recent years. AutoRL methods aim to optimize the hyperparameter selection so that practitioners can avoid time-consuming hyperparameter searches. AutoRL methods also seek to minimize the number of required samples to achieve reasonable performances so that RL algorithms can be used in applications where a limited number of interactions with the environment is possible. AutoRL is still in its early stage of development, and most existing methods adapt techniques that have been shown to be effective for automated Machine Learning (AutoML) (Hutter et al., 2019; Falkner et al., 2018). However, RL brings additional challenges that have been overlooked till now, despite notoriously requiring special care (Igl et al., 2021). For example, due to the highly non-stationary nature of RL, it is unlikely that a static selection of hyperparameters works optimally for a given problem and algorithm (Mohan et al., 2023). This has led to the development of several techniques for adapting the training process online to prevent issues like local minima (Nikishin et al., 2022; Sokar et al., 2023), lack of exploration (Klink et al., 2020), and catastrophic forgetting (Kirkpatrick et al., 2017). In this work, we focus on the line of work in the AutoRL landscape that aims at dynamically changing the hyperparameters during training to achieve good performances from scratch as opposed to approaches that aim at finding promising static hyperparameters to use in a sub-sequent training.

---

[*]Correspondance to: theo.vincent@dfki.de

Q-Network
$\Gamma\bar{Q}_0 \rightarrow Q_1 | \Gamma\bar{Q}_1 \rightarrow Q_2 | \Gamma\bar{Q}_2 \rightarrow \cdots$
$\Gamma\bar{Q}_0 \rightarrow Q_1 | \Gamma\bar{Q}_1 \rightarrow Q_2 | \Gamma\bar{Q}_2 \rightarrow \cdots$
$\Gamma\bar{Q}_0 \rightarrow Q_1 | \Gamma\bar{Q}_1 \rightarrow Q_2 | \Gamma\bar{Q}_2 \rightarrow \cdots$

Adaptive Q-Network (ours)

Figure 1: **Left:** Each line represents a training of *Q-Network* (QN) with different hyperparameters. **Right:** At the $i^{\text{th}}$ target update, *Adaptive Q-Network* (AdaQN) selects the network $Q_i$ (highlighted with a crown) that is the closest to the previous target $\Gamma\bar{Q}_{i-1}$, where $\Gamma$ is the Bellman operator.

We introduce a novel approach for AutoRL to improve the effectiveness of learning algorithms by coping with the non-stationarities of the RL optimization procedure. Our investigation stems from the intuition that the effectiveness of each hyperparameter selection changes dynamically after each training update. Building upon this observation, we propose to *adaptively* select the best-performing hyperparameters configuration to carry out Bellman updates. To highlight this idea, we call our approach *Adaptive Q-Network* (AdaQN). In practice, AdaQN uses an ensemble of $Q$-functions, trained with different hyperparameters, and selects the one with the smallest approximation error (Schaul et al., 2015; D'Eramo & Chalvatzaki, 2022) to be used as a shared target for each Bellman update. By doing this, our method considers several sets of hyperparameters at once each time the stationarity of the optimization problem breaks, i.e., at each target update (Figure 1). In the following, we provide a theoretical motivation for our selection mechanism and then, we empirically validate AdaQN in MuJoCo control problems and Atari 2600 games, showing that it has several advantages over carrying out individual runs with static hyperparameters. Importantly, by trading off different hyperparameters configurations dynamically, AdaQN not only can match the performance of the best hyperparameter selection at no cost of additional samples, but can also reach superior overall performance than any individual run. Finally, we compare AdaQN against strong AutoRL baselines and evince AdaQN's superiority in finding good hyperparameters in a sample-efficient manner.

## 2 PRELIMINARIES

We consider discounted Markov decision processes (MDPs) defined as $\mathcal{M} = \langle \mathcal{S}, \mathcal{A}, \mathcal{P}, \mathcal{R}, \gamma \rangle$, where $\mathcal{S}$ and $\mathcal{A}$ are measurable state and action spaces, $\mathcal{P} : \mathcal{S} \times \mathcal{A} \rightarrow \Delta(\mathcal{S})^1$ is a transition kernel, $\mathcal{R} : \mathcal{S} \times \mathcal{A} \rightarrow \mathbb{R}$ is a reward function, and $\gamma \in [0, 1)$ is a discount factor (Puterman, 1990). A policy is a function $\pi : \mathcal{S} \rightarrow \Delta(\mathcal{A})$, inducing an action-value function $Q^\pi(s, a) \triangleq \mathbb{E}_\pi \left[ \sum_{t=0}^\infty \gamma^t \mathcal{R}(s_t, a_t) | s_0 = s, a_0 = a \right]$ that gives the expected discounted cumulative return executing action $a$ in state $s$, following policy $\pi$ thereafter. The objective is to find an optimal policy $\pi^* = \arg\max_\pi V^\pi(\,\cdot\,)$, where $V^\pi(\,\cdot\,) = \mathbb{E}_{a \sim \pi(\,\cdot\,)}[Q^\pi(\,\cdot\,, a)]$. Approximate value iteration (AVI) and approximate policy iteration (API) are two paradigms to tackle this problem (Sutton & Barto, 1998). While AVI aims at estimating the optimal action-value function $Q^*$, i.e., the action-value function of the optimal policy, API is a two-step procedure that alternates between Approximate policy evaluation (APE), a method to evaluate the action-value function $Q^\pi$ of the current policy $\pi$, and policy improvement, which updates the current policy by taking greedy actions on $Q^\pi$.

Both AVI and APE aim to solve a Bellman equation, whose solution is $Q^*$ for AVI and $Q^\pi$ for APE. Those solutions are the fixed point of a Bellman operator $\Gamma$, where $\Gamma$ is the optimal Bellman operator $\Gamma^*$ for AVI, and the Bellman operator $\Gamma^\pi$ for APE. For a $Q$-function $Q$, a state $s$ and an action $a$, $\Gamma^* Q(s, a) = r + \gamma \mathbb{E}_{s' \sim \mathcal{P}(s,a)}[\max_{a'} Q(s', a')]$ and $\Gamma^\pi Q(s, a) = r + \gamma \mathbb{E}_{s' \sim \mathcal{P}(s,a), a' \sim \pi(s')}[Q(s', a')]$. $\Gamma^*$ and $\Gamma^\pi$ are $\gamma$-contraction mapping in the infinite norm. Because of this, these two methods repeatedly apply their respective Bellman operator, starting from a random $Q$-function. The fixed point theorem ensures that each iteration is closer to the fixed point of the respective Bellman equation. Hence, the more Bellman iterations are performed, the closer the $Q$-function will be to the desired one. In model-free RL, $\Gamma^*$ and $\Gamma^\pi$ are approximated by their empirical version that we note $\hat{\Gamma}$ without distinguishing between AVI and APE since the nature of $\hat{\Gamma}$ can be understood from the context. We denote $\Theta$, the space of $Q$-functions parameters. In common $Q$-Network (QN) approaches, given a *fixed* vector of parameters $\bar{\theta} \in \Theta$, another vector of parameters $\theta$ is learned to minimize

$$\mathcal{L}_{\text{QN}}(\theta | \bar{\theta}, s, a, r, s') = (\hat{\Gamma} Q_{\bar{\theta}}(s, a) - Q_\theta(s, a))^2 \qquad (1)$$

---

[1]$\Delta(\mathcal{X})$ denotes the set of probability measures over a set $\mathcal{X}$.

for a sample $(s, a, r, s')$. $Q_{\bar{\theta}}$ is usually called the target $Q$-function because it is used to compute the target $\hat{\Gamma}Q_{\bar{\theta}}$ while $Q_\theta$ is called the online $Q$-function. We refer to this step as the *projection step* since the Bellman iteration $\Gamma Q_{\bar{\theta}}$ is projected back into the space of $Q$-functions that can be represented by our function approximations. The target parameters $\bar{\theta}$ are regularly updated to the online parameters $\theta$ so that the following Bellman iterations are learned. This step is called the *target update*. We note $\bar{\theta}_i$ the target parameters after the $i^{\text{th}}$ target update. In this work, at each target update $i$, we aim at finding the best hyperparameters $\zeta_i$ characterizing the learnable parameters $\theta_i$, such that

$$\zeta_i = \arg\max_\zeta J(\theta_i) \text{ s.t. } \theta_i \in \arg\min_\theta f(\theta; \zeta, \bar{\theta}_{i-1}), \tag{2}$$

where $J$ is the cumulative discounted return and $f$ is the objective function of the learning process.

## 3   RELATED WORK

Methods for AVI or APE (Mnih et al., 2015; Dabney et al., 2018; Haarnoja et al., 2018; Bhatt et al., 2024) are highly dependent on hyperparameters (Henderson et al., 2018; Andrychowicz et al., 2020; Engstrom et al., 2019). AutoRL tackles this limitation by searching for effective hyperparameters.

We follow the categorization of AutoRL methods presented in Parker-Holder et al. (2022) to position our work in the AutoRL landscape. Contrary to AdaQN, many approaches consider optimizing the hyperparameters through multiple trials. A classic approach is to cover the search space with a grid search or a random search (Hutter et al., 2019; Bergstra & Bengio, 2012). Some other methods use Bayesian optimization to guide the search in the space of hyperparameters (Chen et al., 2018; Falkner et al., 2018; Nguyen et al., 2020; Shala et al., 2022), leveraging the information collected from individual trials. Those methods tackle the problem of finding the best set of hyperparameters but do not consider changing the hyperparameters during a single trial, which would be more appropriate for handling the non-stationarities inherent to the RL problem. To this end, evolutionary approaches have been developed (Stanley et al., 2009; Jaderberg et al., 2019; Awad et al., 2021), where the best elements of a population of agents undergo genetic modifications during training. As an example, in SEARL (Franke et al., 2021), a population of agents is first evaluated in the environment. Then, the best-performing agents are selected, and a mutation process is performed to form the next generation of agents. Finally, the samples collected during evaluation are taken from a shared replay buffer to train each individual agent, and the loop repeats. Even if selecting agents based on their performance in the environment seems reasonable, we argue that it hinders sample-efficiency since exploration in the space of hyperparameters might lead to poorly performing agents. Moreover, if poorly performing agents are evaluated, low-quality samples will then be stored in the replay buffer and used later for training, which could lead to long-term negative effects. This is why we propose to base the selection mechanism on the approximation error, which does not require additional interaction with the environment. Other methods have considered evaluating the $Q$-functions offline (Tang & Wiens, 2021). Most approaches consider the Bellman error $\|\Gamma Q_i - Q_i\|$ instead of the approximation error $\|\Gamma Q_{i-1} - Q_i\|$, where $Q_i$ is the $Q$-function obtained after $i$ Bellman iterations (Farahmand & Szepesvári, 2011; Zitovsky et al., 2023). However, this approach considers the empirical estimate of the Bellman error which is a biased estimate of the true Bellman error (Baird, 1995). Lee et al. (2022) propose to rely on the approximation error to choose between two different classes of functions in an algorithm called ModBE. AdaQN differs from ModBE since ModBE does not adapt the hyperparameters at each Bellman iteration but instead performs several training steps before comparing two classes of functions. Furthermore, the use of the approximation error can be theoretically justified by leveraging Theorem 3.4 from Farahmand (2011), as shown in Section 4.

Meta-gradient reinforcement learning (MGRL) also optimizes the hyperparameters during training (Finn et al., 2017; Zahavy et al., 2020; Flennerhag et al., 2021). However, while MGRL focuses on tuning the target to ease the optimization process, AdaQN uses diverse online networks to overcome optimization hurdles. This shift in paradigm leads AdaQN to avoid some of MGRL's shortcomings: AdaQN can work with discrete hyperparameters (e.g., optimizer, activation function, losses, or neural architecture), only requires first-order approximation, and does not suffer from non-stationarities when hyperparameters are updated since each online network is trained with respect to its own hyperparameters (Xu et al., 2018). Finally, AdaQN fits best in the "Blackbox Online Tuning" cluster even if most methods in this cluster focus on the behavioral policy (Schaul et al., 2019; Badia et al., 2020) or build the hyperparameters as functions of the state of the environment (Sutton & Singh, 1994; White & White, 2016). For example, Riquelme et al. (2019) develop a method called

adaptive TD that selects between the TD-update or the Monte-Carlo update depending on whether the TD-update belongs to a confidence interval computed from several Monte-Carlo estimates.

## 4    ADAPTIVE TEMPORAL-DIFFERENCE TARGET SELECTION

Notoriously, when the AVI/APE projection step described in Section 2 is carried out for non-linear function approximation together with off-policy sampling, convergence to the fixed point is no longer guaranteed. Nevertheless, well-established theoretical results in AVI relate the approximation error $\|\Gamma Q_{\bar{\theta}_{i-1}} - Q_{\bar{\theta}_i}\|$ at each target update $i$ to the performance loss $\|Q^* - Q^{\pi_N}\|$, i.e., the distance between the $Q$-function at timestep $N$ during the training and the optimal one $Q^*$. For a fixed number of Bellman iterations $N$, Theorem 3.4 from Farahmand (2011), stated in Appendix A, demonstrates that the performance loss is upper bounded by a term that decreases when the sum of approximation errors $\sum_{i=1}^{N} \|\Gamma Q_{\bar{\theta}_{i-1}} - Q_{\bar{\theta}_i}\|$ decreases, i.e., every projection step improves. Hence, minimizing the sum of approximation errors is crucial to get closer to the fixed point. However, due to the complexity of the landscape of the empirical loss $\mathcal{L}_{\text{QN}}$ (Mohan et al., 2023), using *fixed* hyperparameters to minimize each approximation error $\|\Gamma Q_{\bar{\theta}_{i-1}} - Q_{\bar{\theta}_i}\|$ is unlikely to be effective and can, in the worst case, lead to local minima (Nikishin et al., 2022; Sokar et al., 2023) or diverging behaviors (Baird, 1995).

In this work, we tackle this problem by introducing a method that can seamlessly handle multiple hyperparameters in a *single* AVI/APE training procedure to speed up the reduction of the performance loss described above, compared to a fixed hyperparameter selection. This method, which we call Adaptive $Q$-Network (AdaQN), learns several online networks trained with different hyperparameters. Crucially, to cope with the non-stationarity of the RL procedure, at each target update, the online network with the lowest approximation error and thus the lowest performance loss, is selected as a shared target network used to train all the online networks. This results in moving all the networks of the ensemble towards the one that is currently the closest to the fixed point, thus benefiting from the heterogeneous ensemble to speed up learning.

We denote $(\theta^k)_{k=1}^{K}$ the parameters of the online networks and $\bar{\theta}_i$ the parameters of the shared target network after the $i^{\text{th}}$ target update. To minimize the $i^{\text{th}}$ approximation error, the optimal choice for $\bar{\theta}_i$ given $\bar{\theta}_{i-1}$, is the index of the closest online network from the target $\Gamma Q_{\bar{\theta}_{i-1}}$:

$$\bar{\theta}_i \leftarrow \theta^\psi \text{ where } \psi = \operatorname*{arg\,min}_{k \in \{1,\dots,K\}} \|\Gamma Q_{\bar{\theta}_{i-1}} - Q_{\theta^k}\|_{2,\nu}^2, \tag{3}$$

where $\nu$ is the distribution of state-action pairs in the replay buffer. Note that Equation (3) contains the true Bellman operator $\Gamma$ which is not known. Instead, we propose to rely on the value of the empirical loss to select the next target network:

$$\bar{\theta}_i \leftarrow \theta^\psi \text{ where } \psi = \operatorname*{arg\,min}_{k \in \{1,\dots,K\}} \sum_{(s,a,r,s') \in \mathcal{D}} \mathcal{L}_{\text{QN}}(\theta^k | \bar{\theta}_{i-1}, s, a, r, s'), \tag{4}$$

where $\mathcal{D}$ is the replay buffer. Theorem 4.1 shows under which condition the selected index in Equation (4) is the same as the one in Equation (3). Importantly, this condition is valid when the dataset is infinite and the samples are generated from the true distribution. Indeed, the empirical Bellman operator is unbiased w.r.t. the true Bellman operator. The proof, inspired by the bias-variance trade-off in supervised learning, can be found in Appendix A.

**Theorem 4.1.** *Let $(\theta^k)_{k=1}^{K} \in \Theta^K$ and $\bar{\theta} \in \Theta$ be vectors of parameters representing $K+1$ Q-functions. Let $\mathcal{D} = \{(s, a, r, s')\}$ be a set of samples. Let $\nu$ be the distribution represented by the state-action pairs present in $\mathcal{D}$. We note $\mathcal{D}_{s,a} = \{(r, s') | (s, a, r, s') \in \mathcal{D}\}, \forall (s, a) \in \mathcal{D}$.*

*If for every state-action pair $(s, a) \in \mathcal{D}$, $\mathbb{E}_{(r,s') \sim \mathcal{D}_{s,a}} \left[ \hat{\Gamma} Q_{\bar{\theta}}(s, a) \right] = \Gamma Q_{\bar{\theta}}(s, a)$, then we have*

$$\operatorname*{arg\,min}_{k \in \{1,\dots,K\}} \|\Gamma Q_{\bar{\theta}} - Q_{\theta^k}\|_{2,\nu}^2 = \operatorname*{arg\,min}_{k \in \{1,\dots,K\}} \sum_{(s,a,r,s') \in \mathcal{D}} \mathcal{L}_{QN}(\theta^k | \bar{\theta}, s, a, r, s').$$

Key to our approach is that each vector of parameters $\theta^k$ can be trained with a different optimizer, learning rate, architecture, activation function, or any other hyperparameter that only affects its training. By cleverly selecting the next target network from a set of diverse online networks, AdaQN has

---

**Algorithm 1** Adaptive Deep $Q$-Network (AdaDQN). Modifications to DQN are marked in purple.

1: Initialize $K$ online parameters $(\theta^k)_{k=1}^K$, and an empty replay buffer $\mathcal{D}$. Set $\psi = 0$ and $\bar{\theta} \leftarrow \theta^\psi$ the target parameters. Set the cumulative losses $L_k = 0$, for $k = 1, \ldots, K$.
2: **repeat**
3:      Set $\psi^b \sim U\{1, \ldots, K\}$ w.p. $\epsilon_b$ and $\psi^b = \psi$ w.p. $1 - \epsilon_b$.
4:      Take action $a_t \sim \epsilon\text{-greedy}(Q_{\theta^{\psi^b}}(s_t, \cdot))$; Observe reward $r_t$, next state $s_{t+1}$.
5:      Update $\mathcal{D} \leftarrow \mathcal{D} \bigcup \{(s_t, a_t, r_t, s_{t+1})\}$.
6:      **every** $G$ **steps**
7:          Sample a mini-batch $\mathcal{B} = \{(s, a, r, s')\}$ from $\mathcal{D}$.
8:          Compute the *shared* target $y \leftarrow r + \gamma \max_{a'} Q_{\bar{\theta}}(s', a')$.
9:          **for** $k = 1, ..., K$ **do**
10:              Compute the loss w.r.t $\theta^k$, $\mathcal{L}_{\mathrm{QN}}^k = \sum_{(s,a,r,s') \in \mathcal{B}} (y - Q_{\theta^k}(s, a))^2$.
11:              Update $L_k \leftarrow L_k + \mathcal{L}_{\mathrm{QN}}^k$.
12:              Update $\theta^k$ using its *specific* optimizer and learning rate from $\nabla_{\theta^k} \mathcal{L}_{\mathrm{QN}}^k$.
13:      **every** $T$ **steps**
14:          Update $\psi \leftarrow \arg\min_k L_k$; $L_k \leftarrow 0$, for $k \in \{1, \ldots, K\}$.
15:          Update the target network with $\bar{\theta} \leftarrow \theta^\psi$.

---

a higher chance of overcoming the typical challenges of optimization presented earlier. Importantly, we prove that AdaQN is guaranteed to converge to the optimal action-value function with probability 1 in a tabular setting in Appendix A.1.

### 4.1 ALGORITHMIC IMPLEMENTATION

Multiple algorithms can be derived from our formulation. Algorithm 1 shows an adaptive version of Deep $Q$-Network (DQN, Mnih et al. (2015)) that we call Adaptive Deep $Q$-Network (AdaDQN). Similarly, Adaptive Soft Actor-Critic (AdaSAC), presented in Algorithm 2, is an adaptive version of Soft Actor-Critic (SAC, Haarnoja et al. (2018)). In SAC, the target network is updated using Polyak averaging (Lillicrap et al., 2015). Therefore, we consider $K$ target networks $(\bar{\theta}^k)_{k=1}^K$. Each target network $\bar{\theta}^k$ is updated from its respective online network $\theta^k$, as shown in Line 11 of Algorithm 2. However, each online network is trained w.r.t. a *shared* target chosen from a single target that comes from the set of $K$ target networks. Similarly to the strategy presented in Equation (4), after the $i^{\text{th}}$ target update, the shared target network is chosen as

$$\bar{\theta}_i \leftarrow \bar{\theta}^\psi \text{ where } \psi = \arg\min_{k \in \{1, \ldots, K\}} \sum_{(s,a,r,s') \in \mathcal{D}} \mathcal{L}_{\mathrm{QN}}(\theta^k | \bar{\theta}_{i-1}, s, a, r, s'). \quad (5)$$

Multiple online networks enable to choose which network to use to sample actions in a value-based setting (see Line 3 of Algorithm 1) and which network is selected to train the actor in an actor-critic setting (see Line 13 of Algorithm 2). This choice is related to the behavioral policy, thus, we choose the same strategy in both settings. Intuitively, the optimal choice would be to pick the best-performing network; however, this information is not available. More importantly, only exploring using the best-performing network could lead the other online networks to learn passively (Section 5.1). This would make the performances of the other networks drop (Ostrovski et al., 2021), making them useless for the rest of the training. Inspired by $\epsilon$-greedy policies commonly used in RL for exploration, we select a random network with probability $\epsilon_b$ and select the online network corresponding to the selected target network (as a proxy for the best-performing network) with probability $1 - \epsilon_b$. We use a linear decaying schedule for $\epsilon_b$. Finally, at each target update, the running loss accumulated after each gradient step is used instead of the loss computed over the entire dataset.

## 5 EXPERIMENTS

We first evaluate our method when hyperparameters are selected from a *finite* space. This setting is relevant when the RL practitioner has a good understanding of most hyperparameters and only hesitates among a few ones. We compare AdaQN to an exhaustive grid search where both approaches have access to the same sets of hyperparameters. This allows us to see how AdaQN performs compared to the best static set of hyperparameters. Additionally, we focus on sample efficiency; thus, *the environment steps used for tuning the hyperparameters are also accounted for when reporting the results.* This is why, for reporting the grid search results, we multiply the number of interactions

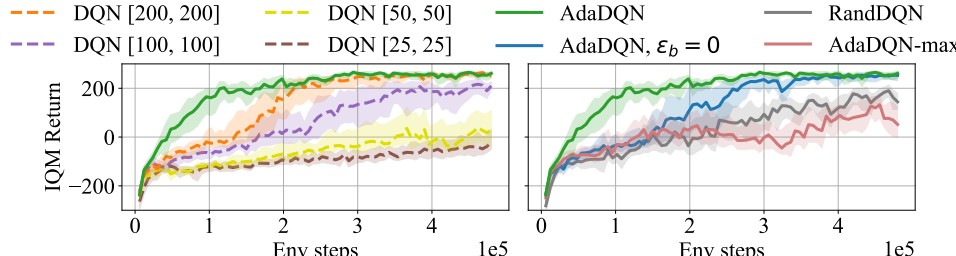

Figure 2: On-the-fly architecture selection on **Lunar Lander**. All architectures contain two hidden layers. The number of neurons in each layer is indicated in the legend. **Left:** AdaDQN yields a better AUC than every DQN run. **Right:** Ablation on the behavioral policy and on the strategy to select the target network used to compute the target. Each version of AdaDQN uses the 4 presented architectures. The strategy presented in Equation (4) outperforms the other considered strategies.

of the best achieved performance by the number of individual trials, as advocated by Franke et al. (2021). We also report the average performance over the individual runs as the performance that a random search would obtain in expectation. Then, we evaluate our method in a second setting for which the hyperparameter space is *infinite*. This setting corresponds to the more general case where the RL practitioner has to select hyperparameters for a new environment, having limited intuitions for most hyperparameters. We compare AdaQN to SEARL (Franke et al. (2021) presented in Section 3), which is specifically designed for this setting, DEHB (Awad et al., 2021), which is a black box optimizer that has been shown to be effective for RL (Eimer et al., 2023), and random search, which remains a very challenging baseline for hyperparameter optimization. In both settings, the remaining hyperparameters are kept unchanged. Appendix B describes all the hyperparameters used for the experiments along with the description of the environment setups and evaluation protocols. The code is based on the Stable Baselines implementation (Raffin et al., 2021) and Dopamine RL (Castro et al., 2018)[2].

**Performance metric.** As recommended by Agarwal et al. (2021), we show the interquartile mean (IQM) along with shaded regions showing pointwise $95\%$ percentile stratified bootstrap confidence intervals. IQM trades off the mean and the median where the tail of the score distribution is removed on both sides to consider only $50\%$ of the runs. We argue that the final score is not enough to properly compare RL algorithms since methods that show higher initial performances are better suited for real-world experiments compared to methods that only show higher performances later during training. This is why we also analyze the performances with the Area Under the Curve (AUC) that computes the integral of the IQM along the training. We also report the worst-performing seed to analyze the robustness of AdaQN w.r.t. stochasticity and highlight AdaQN's stability against hyperparameter changes. We use 20 seeds for Lunar Lander (Brockman et al., 2016), 9 seeds for MuJoCo (Todorov et al., 2012), and 5 seeds for Atari (Bellemare et al., 2013).

## 5.1 A PROOF OF CONCEPT

We consider 4 different architectures for AdaDQN and compare their performances to the individual runs in Figure 2 (left). The 4 architectures are composed of two hidden layers, with the number of neurons indicated in the legend. Interestingly, AdaDQN outperforms the best individual architecture, meaning that by selecting different architectures during the training, AdaDQN better copes with the non-stationarities of the optimization procedure. Figure 2 (right) shows the performances for AdaDQN along with a version of AdaQN where $\epsilon_b = 0$ during the training (AdaDQN $\epsilon_b = 0$), i.e. actions are always sampled for the online network that was chosen for computing the target. This variant of AdaQN performs similarly to the best individual run but underperforms compared to AdaDQN because it suffers from passive learning as explained in Section 4.1. Additionally, we evaluate a version where the target network is selected randomly from the set of online networks, calling this variant RandDQN. This version is similar to the way the target is computed in REDQ (Chen et al., 2020). While sampling uniformly from similar agents yields better performance, as shown in Chen et al. (2020), this strategy suffers when one agent is not performing well. In our case, DQN [25, 25] harms the overall performance. Finally, taking the maximum instead of the minimum to select the target (AdaDQN-max) performs as badly as the worst available agent (i.e., DQN [25, 25]).

---

[2]The code is available in the supplementary material and will be made open source upon acceptance.

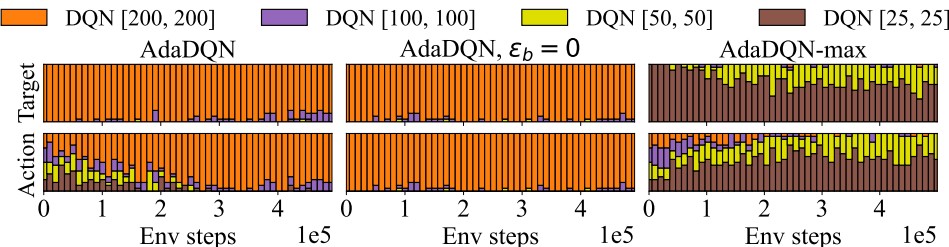

Figure 3: Distribution of the selected hyperparameters for the target network (top) and for the behavioral policy (bottom) across all seeds. **Left:** AdaDQN mainly selects the hyperparameter that performs best when evaluated individually. **Middle:** AdaDQN with $\epsilon_b = 0$ also focuses on the best individual architecture but does not use all available networks for sampling actions, which lowers its performance. **Right:** AdaDQN-max is a version of AdaDQN where the minimum operator is replaced by the maximum operator for selecting the following target network.

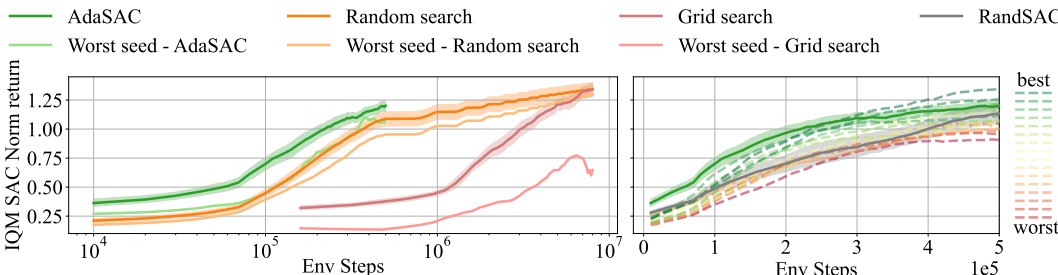

Figure 4: On-the-fly hyperparameter selection on **MuJoCo**. The 16 sets of hyperparameters are the elements of the Cartesian product between the learning rates $\{0.0005, 0.001\}$, the optimizers $\{\text{Adam, RMSProp}\}$, the critic's architectures $\{[256, 256], [512, 512]\}$ and the activation functions $\{\text{ReLU, Sigmoid}\}$. **Left:** AdaSAC is more sample-efficient than random search and grid search. **Right:** AdaSAC yields a better AUC than every SAC run while having a greater final score than 13 out of 16 SAC runs. The shading of the dashed lines indicates their ranking for the AUC metric.

In Figure 3 (top), we analyze the distribution of the selected hyperparameters for the target network across all seeds along the training. Importantly, AdaDQN does not always select the architecture that performs the best when trained individually, as the network DQN $[100, 100]$ is sometimes chosen instead of DQN $[200, 200]$. On the right, AdaDQN-max mainly selects DQN $[25, 25]$, which explains its poor performance. The gap in performance between AdaDQN and AdaDQN with $\epsilon_b = 0$ can be understood by the bottom row of the figure. It shows the distribution of the selected hyperparameters for sampling the actions. Setting $\epsilon_b = 0$ forces the networks not selected for computing the target to learn passively (the top and bottom bar plots are the same for AdaDQN with $\epsilon_b = 0$). On the contrary, AdaDQN benefits from a better exploration at the beginning of the training (the bottom bar plot is more diverse than the top one at the beginning of the training).

## 5.2 CONTINUOUS CONTROL: MUJOCO ENVIRONMENTS

We evaluate AdaSAC for a wider choice of hyperparameters on 6 MuJoCo environments. We consider selecting from 16 sets of hyperparameters. Those sets form the Cartesian product between the learning rates $\{0.0005, 0.001\}$, the optimizers $\{\text{Adam, RMSProp}\}$, the critic's architectures $\{[256, 256], [512, 512]\}$ and the activation functions $\{\text{ReLU, Sigmoid}\}$. This sets $K$ to $2^4 = 16$. The values of the hyperparameters were chosen to be representative of common choices made when tuning by hand a SAC agent. In Figure 4 (left), we compare AdaSAC with a grid search performed on the 16 set of hyperparameters. AdaSAC is an order of magnitude more sample efficient than grid search and outperforms random search all along the training. Notably, AdaSAC's worst-performing seed performs on par with the random search approach while greatly outperforming the worst-performing seed of the grid search. In Figure 4 (right), we show AdaSAC's performance along with the individual performance of each set of hyperparameters. AdaSAC yields the highest

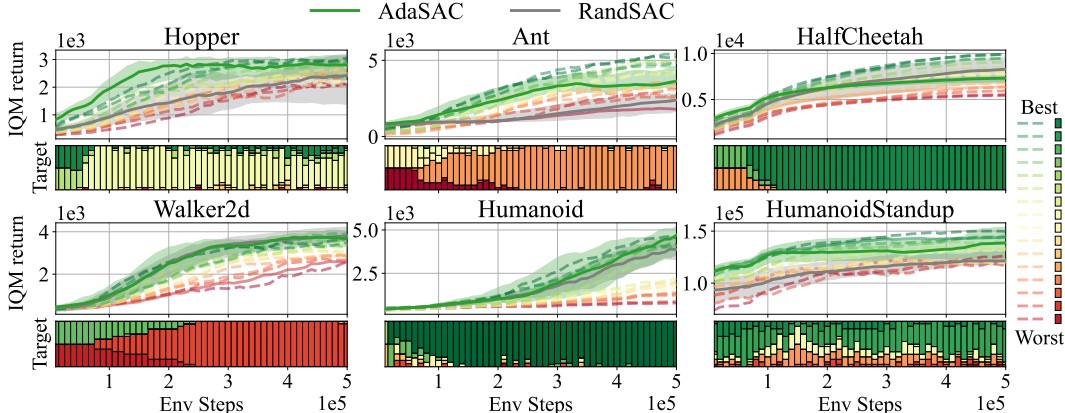

Figure 5: Per environment IQM return when AdaSAC and RandSAC select from the 16 sets of hyperparameters described in Figure 4. Below each performance plot, a bar plot presents the distribution of hyperparameters selected for the target network across all seeds. AdaSAC outperforms RandSAC and most individual run by designing non-trivial hyperparameter schedules.

AUC metric while outperforming 13 out of the 16 individual runs in terms of final performance. Interestingly, AdaSAC reaches the performance of vanilla SAC in less than half of the samples. We also show the performance of RandSAC, for which a random target network is selected to compute the target instead of following the strategy presented in Equation (5). For clarity, the labels of the 16 individual runs are not shown. They are available in Figure 11 of Appendix D.1.

We now analyze which set of hyperparameters AdaSAC selects to compute the target. For each environment, we show in Figure 5 the performances of each hyperparameter when trained individually, AdaSAC and RandSAC performances. Below each performance plot, we report the distribution of hyperparameters selected to compute the target across all seeds. Overall, the selected networks are changing during the training, which indicates that the loss is not always minimized by the same set of hyperparameters. This supports the idea of selecting the target network based on Equation (4). Furthermore, the selected networks are not the same across the different environments. This shows that AdaSAC designs non-trivial hyperparameter schedules that cannot be handcrafted. On *Humanoid*, selecting the network corresponding to the best-performing agent leads to similar performances to the best-performing agents when trained individually. On *HumanoidStandup*, the selection strategy avoids selecting the worst-performing agents. This is why AdaSAC outperforms RandSAC, which blindly selects agents. *HalfCheetah* is the only environment where RandSAC slightly outperforms AdaSAC. AdaSAC still yields a better final performance than 6 of the individual runs. Finally, on *Hopper* and *Walker2d*, AdaSAC outperforms every individual run for the AUC metric by mainly selecting hyperparameters that are *not* performing well when trained individually. The ability of AdaSAC to adapt the hyperparameters at each target update enables it to better fit the targets, which yields better performances.

**Ablations.** We evaluate AdaSAC against random search and grid search in 4 different scenarios where only one hyperparameter changes. The results presented in Appendices D.2, D.3, D.4, and D.5 show that AdaSAC is robust to variations of the hyperparameters. We first give 4 architectures of different capacities as input. Then, we run AdaSAC with 3 different learning rates, and we also run another study where AdaSAC is given 3 different optimizers. In each case, AdaSAC matches the best-performing individual runs while successfully ignoring the worst-performing sets of hyperparameters. Finally, we evaluate AdaSAC with 3 different activation functions given as input. This time, AdaSAC outperforms 2 out of the 3 individual runs. Interestingly, RandSAC seems to suffer more in that setting by only outperforming the worst-performing individual run. This further demonstrates that the strategy presented in Equation (4) is effective.

## 5.3 VISION-BASED CONTROL: ATARI 2600

We provide an additional experiment on 3 games of the Atari benchmark. Similarly to the ablation study made on the MuJoCo benchmark, we conclude, in light of Figure 6, that AdaDQN performs on

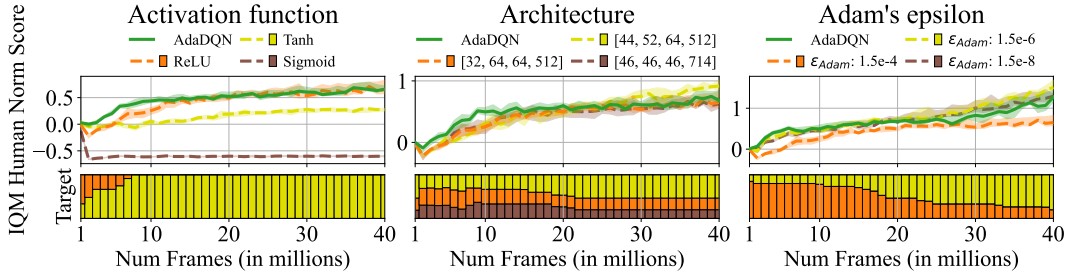

Figure 6: On-the-fly hyperparameter selection on 3 **Atari** games. AdaDQN performs similarly to the best individual hyperparameters when selecting from 3 activation functions (left), 3 architectures (middle), or 3 different values of Adam's epsilon parameter (right). Below each performance plot, a bar plot presents the distribution of the hyperparameters selected for computing the target.

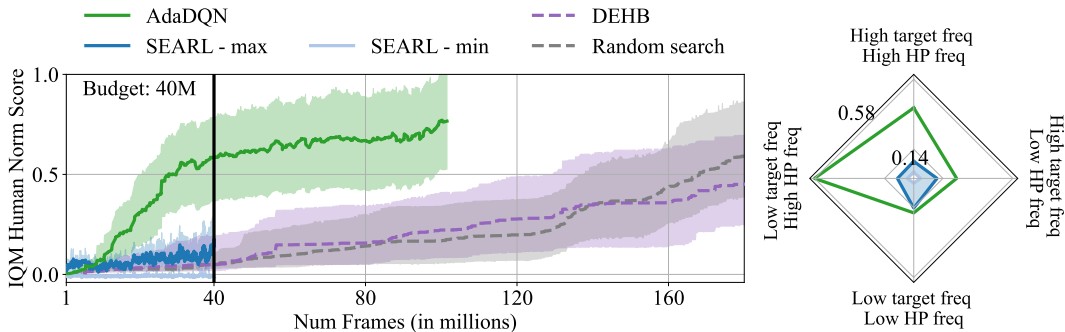

Figure 7: On-the-fly hyperparameter selection on 6 **Atari** games. Each method has to find well-performing hyperparameters in a very challenging hyperparameter space composed of 18 activation functions, 24 optimizers, 4 losses, a wide range of learning rates, and fully parameterizable architectures. **Left:** AdaDQN outperforms SEARL, random search, and grid search in the limited budget of 40M frames and keeps increasing its performance afterward. **Right:** AdaDQN is robust to hyperparameter changes. Each axis of the radar plot shows the performances at 40M frames.

par with the best-performing hyperparameter when the space of hyperparameters is composed of different activation functions (left), architectures (middle) or different values of Adam's epsilon parameter (right). This means that random search and grid search underperform compared to AdaDQN. Importantly, the common hyperparameters used for DQN (in orange) are not the best-performing ones in all settings, which motivates using Auto RL. Below each performance plot, we report the distribution of the hyperparameters selected for computing the target. Remarkably, AdaDQN never selects Sigmoid activations, which perform poorly when evaluated individually. Interestingly, for the ablation study on the architecture and Adam's epsilon parameter, AdaDQN uses all available hyperparameters when they perform equally well. We note that, despite not being often tuned, Adam's epsilon parameter plays a role in yielding high returns as observed in Obando-Ceron et al. (2024). Figure 20 reports the decomposition of Figure 6 into the 3 considered Atari games.

## 5.4 INFINITE HYPERPARAMETER SPACES

We now focus on a setting in which the space of hyperparameters is infinite. We design a wide space of hyperparameters to come closer to the goal of removing the need for prior knowledge about the algorithm and environment. The hyperparameter space, presented in Table 4, comprises 18 activation functions, 24 optimizers, 4 losses, a wide range of learning rates, and fully parameterizable architectures. We let SEARL and AdaDQN train 5 neural networks in parallel. Figure 9 shows how AdaDQN and SEARL differ. To generalize AdaQN to infinite hyperparameter spaces, we replace the RL training step in SEARL's pipeline with the one described in Section 4, and the approximation errors are used as fitness values instead of the online evaluation of each network. The exploitation and exploration steps remain identical. We fix a budget of 40M frames on 6 games of the Atari benchmark. We reduce the budget to 30M frames for random search to favor more individual trials,

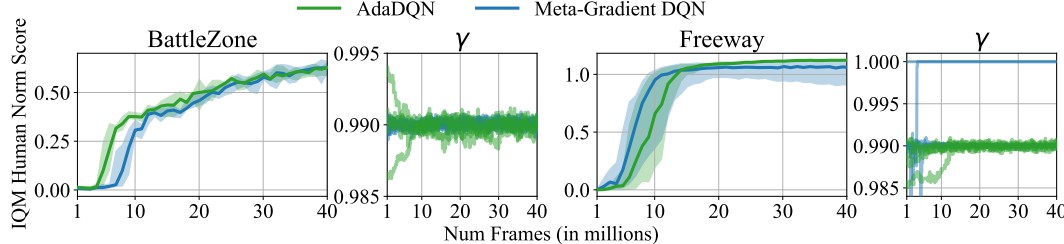

Figure 8: On-the-fly discount factor selection on 2 **Atari** games. AdaDQN and Meta-gradient DQN perform similarly. On the right of each performance plot, we report the evolution of the discount factor selected by AdaDQN and Meta-gradient DQN.

6 in total, as the hyperparameter space is large. Finally, we let DEHB benefit from its multi-fidelity component by allowing a varying budget from 20M to 40M frames. We report the maximum return observed since the beginning of the training for each seed of random search and DEHB. As every network is evaluated in SEARL, we report the maximum and minimum return observed at each evaluation step (SEARL - max and SEARL - min).

In Figure 7 (left), at 40M frames, AdaDQN outperforms SEARL and matches the end performance of random search and DEHB. When trained for longer, AdaDQN continues to increase its performance and outperforms random search and DEHB before reaching 100M frames. We argue that SEARL's low performance is largely explained by the fact that many environment iterations are wasted by poorly performing networks (see SEARL - min), which also affects the other networks as the replay buffer is shared among them. Instead, AdaDQN only uses the networks that have a low training loss. Random search and DEHB usually require many environment interactions to perform well. We believe that having a wide hyperparameter space exacerbates this phenomenon. Figure 7 (right) highlights AdaDQN's robustness w.r.t. its hyperparameters. Notably, AdaDQN reaches a higher IQM return than SEARL in the 4 considered settings at 40M frames. Figure 21 and 22 show the individual game performances. Figure 23 reports the evolution of the hyperparameters during AdaDQN's training for one seed on *Pong*. Interestingly, the performances increase (green background) when the network trained on the Huber loss with ReLU activations and Adam optimizer is selected as target network.

AdaQN can also be used to select the target hyperparameters. We propose to select from different discount factors sampled in $[0.985, 0.995]$. We consider $K = 5$ online networks and compare AdaDQN against Meta-gradient reinforcement learning (MGRL, Xu et al. (2018)). The cross-validated discount factor of MGRL is set at $0.99$, and the same discount factor is considered when selecting the next target network in AdaQN. Figure 8 shows that the two approaches reach similar performances on 2 Atari games. Nevertheless, we recall that AdaQN differs from MGRL in many points (see Section 3). Crucially, it is not limited to continuous and differentiable hyperparameters.

# 6 DISCUSSION AND CONCLUSION

We have introduced Adaptive $Q$-Network (AdaQN), a new method that trains an ensemble of diverse $Q$-networks w.r.t. a shared target, selected as the target network corresponding to the most accurate online network. We demonstrated that AdaQN is theoretically sound and we devised its algorithmic implementation. Our experiments show that AdaQN outperforms competitive AutoRL baselines in terms of sample efficiency, final performance, and robustness.

**Limitations.** Our work focuses on the agent's hyperparameters; thus, environment's properties, e.g., reward, cannot be handled by AdaQN. Nevertheless, an off-the-shelf AutoRL algorithm can be combined with AdaQN to handle these properties. Moreover, since AdaQN considers multiple $Q$-functions in the loss, the training time and memory requirements increase. We provide an analysis in Appendix C to show that this increase remains reasonable compared to the gain in performance and sample-efficiency. We stress that, in theory, the computations can be parallelized so that the adaptive version of an algorithm requires the same amount of time as its original algorithm. The additional "for loop" in Line 9 of Algorithm 1 and the one in Line 7 of Algorithm 2 can be run in parallel as long as enough parallel processing capacity is available, as it is common in modern GPUs. Finally, instead of selecting one $Q$-function for computing the target, future works could consider a mixture of $Q$-functions learned with different hyperparameters as presented in Seyde et al. (2022).

ACKNOWLEDGMENTS

This work was funded by the German Federal Ministry of Education and Research (BMBF) (Project: 01IS22078). This work was also funded by Hessian.ai through the project 'The Third Wave of Artificial Intelligence – 3AI' by the Ministry for Science and Arts of the state of Hessen, by the grant "Einrichtung eines Labors des Deutschen Forschungszentrum für Künstliche Intelligenz (DFKI) an der Technischen Universität Darmstadt", and by the Hessian Ministry of Higher Education, Research, Science and the Arts (HMWK).

CARBON IMPACT

As recommended by Lannelongue & Inouye (2023), we used GreenAlgorithms (Lannelongue et al., 2021) and ML $CO_2$ Impact (Lacoste et al., 2019) to compute the carbon emission related to the production of the electricity used for the computations of our experiments. We only consider the energy used to generate the figures presented in this work and ignore the energy used for preliminary studies. The estimations vary between 1.30 and 1.53 tonnes of $CO_2$ equivalent. As a reminder, the Intergovernmental Panel on Climate Change advocates a carbon budget of 2 tonnes of $CO_2$ equivalent per year per person.

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

# A  THEOREM STATEMENTS AND PROOFS

**Theorem 3.4 from Farahmand (2011).** *Let $N \in \mathbb{N}^*$, and $\rho$, $\nu$ two probability measures on $\mathcal{S} \times \mathcal{A}$. For any sequence $(\bar{\theta}_i)_{i=0}^N \in \Theta^{N+1}$ where $R_\gamma$ depends on the reward function and the discount factor, we have*

$$\|Q^* - Q^{\pi_N}\|_{1,\rho} \leq C_{N,\gamma,R_\gamma} + \inf_{r \in [0,1]} F(r; N, \rho, \gamma) \left( \sum_{i=1}^N \alpha_i^{2r} \|\Gamma^* Q_{\bar{\theta}_{i-1}} - Q_{\bar{\theta}_i}\|_{2,\nu}^2 \right)^{\frac{1}{2}}$$

*where $C_{N,\gamma,R_\gamma}$, $F(r; N, \rho, \gamma)$, and $(\alpha_i)_{i=0}^N$ do not depend on $(\bar{\theta}_i)_{i=0}^N$. $\pi_N$ is a greedy policy computed from $Q_{\theta_N}$.*

**Theorem 4.1.** *Let $(\theta^k)_{k=1}^K \in \Theta^K$ and $\bar{\theta} \in \Theta$ be vectors of parameters representing $K + 1$ Q-functions. Let $\mathcal{D} = \{(s, a, r, s')\}$ be a set of samples. Let $\nu$ be the distribution represented by the state-action pairs present in $\mathcal{D}$. We note $\mathcal{D}_{s,a} = \{(r, s')|(s, a, r, s') \in \mathcal{D}\}, \forall (s, a) \in \mathcal{D}$.*

*If for every state-action pair $(s, a) \in \mathcal{D}$, $\mathbb{E}_{(r,s') \sim \mathcal{D}_{s,a}} \left[ \hat{\Gamma} Q_{\bar{\theta}}(s, a) \right] = \Gamma Q_{\bar{\theta}}(s, a)$, then we have*

$$\arg\min_{k \in \{1,...,K\}} \|\Gamma Q_{\bar{\theta}} - Q_{\theta^k}\|_{2,\nu}^2 = \arg\min_{k \in \{1,...,K\}} \sum_{(s,a,r,s') \in \mathcal{D}} \mathcal{L}_{QN}(\theta^k | \bar{\theta}, s, a, r, s').$$

*Proof.* Let $(\theta^k)_{k=1}^K \in \Theta^K$ and $\bar{\theta} \in \Theta$ be vectors of parameters representing $K + 1$ Q-functions. Let $\mathcal{D} = \{(s, a, r, s')\}$ be a set of samples. Let $\nu$ be the distribution of the state-action pairs present in $\mathcal{D}$. In this proof, we note $\hat{\Gamma}$ as $\hat{\Gamma}_{r,s'}$ to stress its dependency on the reward $r$ and the next state $s'$. For every state-action pair $(s, a)$ in $\mathcal{D}$, we define the set $\mathcal{D}_{s,a} = \{(r, s')|(s, a, r, s') \in \mathcal{D}\}$ and assume that $\mathbb{E}_{(r,s') \sim \mathcal{D}_{s,a}}[\hat{\Gamma}_{r,s'} Q_{\bar{\theta}}(s, a)] = \Gamma Q_{\bar{\theta}}(s, a)$. Additionally, we note $M$ the cardinality of $\mathcal{D}$, $M_{s,a}$ the cardinality of $\mathcal{D}_{s,a}$ and $\mathring{\mathcal{D}}$ the set of unique state-action pairs in $\mathcal{D}$. We take $k \in \{1, \ldots, K\}$ and write

$$\sum_{(s,a,r,s') \in \mathcal{D}} \mathcal{L}_{QN}(\theta^k | \bar{\theta}, s, a, r, s') = \sum_{(s,a,r,s') \in \mathcal{D}} \left( \hat{\Gamma}_{r,s'} Q_{\bar{\theta}}(s, a) - Q_{\theta^k}(s, a) \right)^2$$

$$= \sum_{(s,a,r,s') \in \mathcal{D}} \left( \hat{\Gamma}_{r,s'} Q_{\bar{\theta}}(s, a) - \Gamma Q_{\bar{\theta}}(s, a) + \Gamma Q_{\bar{\theta}}(s, a) - Q_{\theta^k}(s, a) \right)^2$$

$$= \sum_{(s,a,r,s') \in \mathcal{D}} \left( \hat{\Gamma}_{r,s'} Q_{\bar{\theta}}(s, a) - \Gamma Q_{\bar{\theta}}(s, a) \right)^2$$

$$+ \sum_{(s,a,r,s') \in \mathcal{D}} (\Gamma Q_{\bar{\theta}}(s, a) - Q_{\bar{\theta}^k}(s, a))^2$$

$$+ 2 \sum_{(s,a,r,s') \in \mathcal{D}} \left( \hat{\Gamma}_{r,s'} Q_{\bar{\theta}}(s, a) - \Gamma Q_{\bar{\theta}}(s, a) \right) (\Gamma Q_{\bar{\theta}}(s, a) - Q_{\theta^k}(s, a)).$$

The second last equation is obtained by introducing the term $\Gamma Q_{\bar{\theta}}(s, a)$ and removing it. The last equation is obtained by developing the previous squared term. Now, we study each of the three terms:

- $\sum_{(s,a,r,s') \in \mathcal{D}} \left( \hat{\Gamma}_{r,s'} Q_{\bar{\theta}}(s, a) - \Gamma Q_{\bar{\theta}}(s, a) \right)^2$ is independent of $\theta^k$

- $\sum_{(s,a,r,s') \in \mathcal{D}} (\Gamma Q_{\bar{\theta}}(s, a) - Q_{\theta^k}(s, a))^2$ equal to $M \times \|\Gamma Q_{\bar{\theta}} - Q_{\theta^k}\|_{2,\nu}^2$ by definition of $\nu$.

- 

$$\sum_{(s,a,r,s')\in\mathcal{D}} \left(\hat{\Gamma}_{r,s'}Q_{\bar{\theta}}(s,a) - \Gamma Q_{\bar{\theta}}(s,a)\right)\left(\Gamma Q_{\bar{\theta}}(s,a) - Q_{\theta^k}(s,a)\right)$$

$$= \sum_{(s,a)\in\mathring{\mathcal{D}}} \left[\sum_{(r,s')\in\mathcal{D}_{s,a}} \left(\hat{\Gamma}_{r,s'}Q_{\bar{\theta}}(s,a) - \Gamma Q_{\bar{\theta}}(s,a)\right)\left(\Gamma Q_{\bar{\theta}}(s,a) - Q_{\theta^k}(s,a)\right)\right] = 0$$

since, for every $(s,a) \in \mathring{\mathcal{D}}$,

$$\sum_{(r,s')\in\mathcal{D}_{s,a}} \left(\hat{\Gamma}_{r,s'}Q_{\bar{\theta}}(s,a) - \Gamma Q_{\bar{\theta}}(s,a)\right) = M_{s,a}\left(\mathbb{E}_{(r,s')\sim\mathcal{D}_{s,a}}[\hat{\Gamma}_{r,s'}Q_{\bar{\theta}}(s,a)] - \Gamma Q_{\bar{\theta}}(s,a)\right) = 0,$$

the last equality holds from the assumption.

Thus, we have

$$\sum_{(s,a,r,s')\in\mathcal{D}} \mathcal{L}_{\text{QN}}(\theta^k|\bar{\theta},s,a,r,s') = \text{constant w.r.t } \theta^k + M \times ||\Gamma Q_{\bar{\theta}} - Q_{\theta^k}||^2_{2,\nu}$$

This is why

$$\underset{k\in\{1,\dots,K\}}{\arg\min} \sum_{(s,a,r,s')\in\mathcal{D}} \mathcal{L}_{\text{QN}}(\theta^k|\bar{\theta},s,a,r,s') = \underset{k\in\{1,\dots,K\}}{\arg\min} ||\Gamma Q_{\bar{\theta}} - Q_{\theta^k}||^2_{2,\nu}.$$

$\square$

### A.1 CONVERGENCE OF ADAQN

In this section, we consider a tabular version of AdaQN in a finite MDP where the discount factor $\gamma < 1$. Our goal is to prove the convergence of AdaQN to the optimal action-value function by adapting the proof of Theorem 2 in Lan et al. (2020). Without modification, the Generalized $Q$-learning framework (Lan et al., 2020) does not comprise AdaQN. Indeed, while the target computation of the Generalized $Q$-learning framework (Equation (3) in Lan et al. (2020)) only depends on a fixed history length, AdaQN's target computation (Equation (4)) depends on $\psi$, the index of the selected network used for computing the target, which is defined from the entire history of the algorithm. This is why, in the following, we first present the modifications brought to the Generalized $Q$-learning framework before proving the convergence.

We color in blue the modifications brought to the Generalized $Q$-learning framework. Using the same notation as in Equations (15) and (16) in Lan et al. (2020), for a state $s$, an action $a$, and at timestep $t$, the new update rule is

$$Q^i_{s,a}(t+1) \leftarrow Q^i_{s,a}(t) + \alpha^i_{s,a}(t)\left[F_{s,a}(Q(t),\psi(t)) - Q^i_{s,a}(t) + \omega_{s,a}(t)\right], \forall i \in \{1,..,N\}, \quad (6)$$

where $\psi(t) \in \{1,..,N\}$ is $\mathcal{F}(t)$-measurable and

$$\omega_{s,a}(t) = r_{s,a} - \mathbb{E}[r_{s,a}] + \gamma\left(Q^{GQ,\psi(t)}_{f(s,a)}(t) - \mathbb{E}[Q^{GQ,\psi(t)}_{f(s,a)}(t)|\mathcal{F}(t)]\right). \quad (7)$$

We recall that in Lan et al. (2020), $Q \in \mathbb{R}^{mnNK}$ where $m$ is the cardinality of $\mathcal{S}$, $n$ the cardinality of $\mathcal{A}$, $N$ is the population size, and $K$ is the history length where $Q^{i,j}_{s,a}(t)$ is the component corresponding to the $i^{\text{th}}$ element of the population at timestep $t-j$ for state $s$ and action $a$. $\mathcal{F}(t)$ is the $\sigma$-algebra representing the history of the algorithm during the first $t$ iterations. $\alpha^i_{s,a}(t)$ is the step size and $f(s,a)$ is a random successor state starting from state $s$ after executing action $a$. We define, for $i \in \{1,..,N\}, F_{s,a}(Q,i) = \mathbb{E}[r_{s,a}] + \gamma\mathbb{E}[Q^{GQ,i}_{f(s,a)}]$, where $Q^{GQ,i}_s = G(Q_s,i)$.

For AdaQN, $\psi(t)$ represents the index of the selected target at timestep $t$. Following Equation (4), $\psi(t) = \arg\min_i ||\mathbb{E}[r_{s,a}] + \gamma\mathbb{E}[Q^{GQ,\psi(t-1)}_{f(s,a)}] - Q^{i,0}||_2$, with $\psi(0) = 0$, and for $i \in \{1,..,N\}, G(Q_s,i) = \max_a Q^{i,0}_{s,a}$. It is clear that $\psi(t)$ is $\mathcal{F}(t)$-measurable.

We now verify that Assumptions $1, 2, 3, 4,$ and $6$ in Lan et al. (2020), needed to prove the convergence according to Theorem 5 in Lan et al. (2020), hold for AdaQN. Assumption 1 holds as $\forall Q, Q' \in \mathbb{R}^{nNK}, \iota, \iota' \in \{1, .., N\}$, we have $|G(Q, \iota) - G(Q', \iota')| \leq \max_{a,i,j} |Q_a^{i,j} - Q_a'^{i,j}|$. Assumptions $3, 4,$ and $6$ also hold as the modification of the target computation does not influence their validity. We point out that the constraint that $\psi(t)$ is $\mathcal{F}(t)$-measurable is used to verify that $\omega_{s,a}(t)$ is $\mathcal{F}(t+1)$-measurable for all state $s$ and action $a$. This validates Assumption $4$ $(ii)$. Therefore, by defining the step sizes such that Assumption 2 holds, AdaQN converges to the optimal action-value function with probability 1.

## B  ALGORITHMS AND HYPERPARAMETERS

---

**Algorithm 2** Adaptive Soft Actor-Critic (AdaSAC). Modifications to SAC are marked in purple.

---

1: Initialize the policy parameters $\phi$, $K$ online parameters $(\theta^k)_{k=1}^K$, and an empty replay buffer $\mathcal{D}$. For $k = 1, \ldots, K$, set the target parameters $\bar{\theta}^k \leftarrow \theta^k$ and the cumulative losses $L_k = 0$. Set $\psi_1 = 0$ and $\psi_2 = 1$ the indices to be selected for computing the target.
2: **repeat**
3:    Take action $a_t \sim \pi_\phi(\cdot|s_t)$; Observe reward $r_t$, next state $s_{t+1}$; $\mathcal{D} \leftarrow \mathcal{D} \bigcup \{(s_t, a_t, r_t, s_{t+1})\}$.

4:    **for** UTD updates **do**
5:       Sample a mini-batch $\mathcal{B} = \{(s, a, r, s')\}$ from $\mathcal{D}$.
6:       Compute the *shared* target

$$y \leftarrow r + \gamma \left( \min_{k \in \{\psi_1, \psi_2\}} Q_{\bar{\theta}^k}(s', a') - \alpha \log \pi_\phi(a'|s') \right), \text{where } a' \sim \pi_\phi(\cdot|s').$$

7:       **for** $k = 1, ..., K$ **do**
8:          Compute the loss w.r.t $\theta^k$, $\mathcal{L}_{\text{QN}}^k = \sum_{(s,a,r,s') \in \mathcal{B}} (y - Q_{\theta^k}(s, a))^2$.
9:          Update $L_k \leftarrow (1 - \tau)L_k + \tau \mathcal{L}_{\text{QN}}^k$.
10:         Update $\theta^k$ using its *specific* optimizer and learning rate from $\nabla_{\theta^k} \mathcal{L}_{\text{QN}}^k$.
11:         Update the target networks with $\bar{\theta}^k \leftarrow \tau \theta^k + (1 - \tau)\bar{\theta}^k$.
12:      Set $\psi_1$ and $\psi_2$ to be the indexes of the 2 lowest values of $L$.
13:      Set $(\psi_1^b, \psi_2^b) \sim \text{Distinct} U\{1, \ldots, K\}$ w.p $\epsilon_b$ and $(\psi_1^b, \psi_2^b) = (\psi_1, \psi_2)$ w.p $1 - \epsilon_b$.
14:      Update $\phi$ with gradient ascent using the loss

$$\min_{k \in \{\psi_1^b, \psi_2^b\}} Q_{\theta^k}(s, a) - \alpha \log \pi_\phi(a|s), \quad a \sim \pi_\phi(\cdot|s)$$

---

### B.1  EXPERIMENTAL SETUP

All environments are generated from the Gymansium library (Brockman et al., 2016). For the Atari experiments, we build our codebase on Vincent et al. (2025) which mimics the implementation choices made in Dopamine RL (Castro et al., 2018). Dopamine RL follows the training and evaluation protocols recommended by Machado et al. (2018). Namely, we use the *game over* signal to terminate an episode instead of the *life* signal. The input given to the neural network is a concatenation of 4 frames in grayscale of dimension 84 by 84. To get a new frame, we sample 4 frames from the Gym environment configured with no frame-skip, and we apply a max pooling operation on the 2 last grayscale frames. We use sticky actions to make the environment stochastic (with $p = 0.25$). The performance is the one obtained during training. We consider a different behavioral policy for the experiment with an infinite hyperparameter space. Instead of taking an $\epsilon$-greedy policy parameterized by $\epsilon_b$ as explained in Section 4.1, we sample each action from a probability distribution that is inversely proportional to the current loss value. That way, the problem of passive learning is further mitigated since the second or third most accurate networks will have a higher probability of being selected to sample actions. Importantly, when AdaDQN selects between different losses, the comparison is made between the values of the L2 losses even if the backpropagation is performed with another loss. The 3 Atari games in the experiments of Section 5.3 are chosen from the 5 games presented in Aitchison et al. (2023). The 6 Atari games in the experiments of Section 5.4 are chosen from Franke et al. (2021).

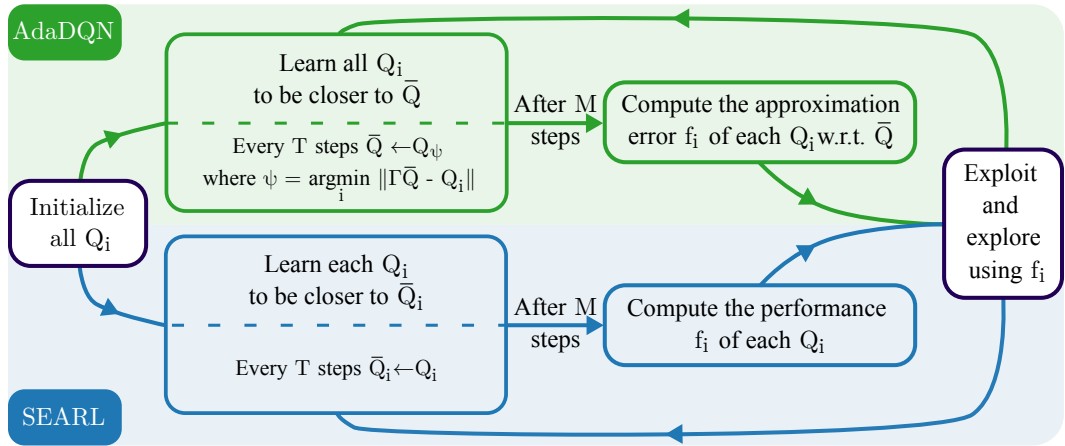

Figure 9: AdaDQN and SEARL pipelines. AdaDQN and SEARL share the same exploitation and exploration strategies. They only differ in the way the $Q$-functions are trained and in the nature of the fitness value $f_i$. In both algorithms, the exploitation and exploration phase is performed every $M$ steps. It is referred to as the hyperparameter frequency (HP freq) in Figures 7 and 22.

**AdaQN - SEARL comparison.** The exploitation strategy of SEARL and AdaDQN is based on a tournament selection with elitism (Miller et al., 1995). The tournament comprises $K - 1$ steps for a population of $K$ networks. At each step, 3 networks are sampled uniformly without replacement, and the network with the best fitness value $f_i$ is qualified to populate the next generation. The last spot is reserved for the network with the best fitness value across the entire population. Then, every network present more than twice in the population undergoes a mutation operation. This is the exploration phase. One mutation operation out of the 5 operations we consider (architecture change, activation function change, loss change, optimizer change, and learning rate change) is chosen with equal probability. When the architecture is changed, one of the following operations is selected $\{\pm 1$ CNN layer, $\pm 4$ CNN channels, $\pm 1$ kernel size, $\pm 1$ CNN stride, $\pm 1$ MLP layer, $\pm 16$ MLP neurons$\}$ with probability $[0.1, \frac{0.4}{3}, \frac{0.4}{3}, \frac{0.4}{3}, 0.1, 0.4]$. Those probabilities are not tuned. They are set by hand so that the number of layers of the CNN and MLP do not change too often. If the hyperparameter goes outside the search space described in Table 4, the value is clipped to the boundary. In every case, the network weights are preserved as much as possible. This means that when the architecture changes, only the new weights that were not present before the mutation operation are generated. We choose to reset the optimizer each time a network undergoes a genetic modification. Instead of an elitism mechanism, we investigated, on Lunar Lander, a truncation mechanism (Jaderberg et al., 2023) where the $20\%$ most accurate networks are selected for the next generation. The performances are similar to the ones obtained with the elitism mechanism. We also tried to reset the weights after each generation. This did not lead to performance improvements. Therefore, we kept the proposed pipeline as close to SEARL as possible.

SEARL evaluates each network's return empirically to estimate its performance. We set the minimum number of environment steps to evaluate each network to $M$ divided by the number of networks in the population so that at least $M$ new samples are collected in the replay buffer. This also synchronizes the frequency at which the hyperparameters are updated with AdaDQN. SEARL network weights are updated every 4 training step, similarly to AdaDQN ($G = 4$ in Table 4).

**AdaQN - Meta-gradient RL comparison.** The experiment comparing AdaDQN to Meta-gradient DQN is made with the default hyperparameters of DQN. Each network of AdaDQN is trained with a different discount factor, and the comparison between online networks is made by computing the approximation error parameterized by a cross-validated discount factor $\bar{\gamma}$. The optimizer used for the outer loss of Meta-gradient DQN is Adam with a learning rate of $0.001$ as in the original paper (Xu et al., 2018). Similarly to Xu et al. (2018), we set the size of the discount factor's embedding space to 16.

B.2 EVALUATION PROTOCOL

The reported performance of the individual runs of DQN and SAC as well as AdaDQN, AdaDQN with $\epsilon_b = 0$, RandDQN, AdaDQN-max, AdaSAC, RandSAC, and Meta-Gradient DQN are the

raw performance obtained during training aggregated over seeds and environments. The reported performances of the other methods require some post-processing steps to account for the way they operate. In the following, for a hyperparameter $k$ and after $t$ environment steps, we note $s_k^{i,j}(t)$ the return obtained during training for a task $i$ and a seed $j$. Additionally, we note $K$ the number of hyperparameters considered for the specific setting. For the *finite* search space setting, we first compute all possible individual runs (Figure 4 (right)) and, in Figure 4 (left), we derive:

- **Grid search** as the IQM of the best individual hyperparameters across each task at each timestep $t$. We scale horizontally the IQM to the right by the number of individual runs to account for the environment steps taken by all individual runs. Indeed, we first identify the hyperparameter $k_i^*(t)$ with the highest IQM for each task $i$ at each timestep $t$: $k_i^*(t) = \arg\max \text{IQM}_j(s_k^{i,j}(t))$. Then, we report $\text{IQM}_{\text{grid search}}(t \times K) = \text{IQM}_{i,j}(s_{k_i^*(t)}^{i,j}(t))$.

- **Random search** as the empirical average over all individual runs. We approximate the empirical average using Monte Carlo sampling for the possible sequence of hyperparameters. This corresponds to the expected performance of what a random search would look like if only one individual trial were attempted sequentially.

In the *infinite* search space setting (Figure 7 (left)), we report the following algorithm:

- **Random search & DEHB**: to make random search and DEHB more competitive in this setting, we keep the samples collected by the previous individual trials when a new trial starts. This means that, for each task and each seed, random search and DEHB generate a sequence of returns over 180M environment interactions. From this sequence, we compute a running maximum over timesteps. Finally, we compute the IQM over seeds and tasks. In detail, the score used for computing the IQM is $\max_{t' \leq t} s_k^{i,j}(t')$.

- **SEARL**: SEARL has the specificity of periodically evaluating every $Q$-network of its population. Each time every $Q$-network is evaluated, we use the best return to compute the IQM of SEARL-max $(\max_{k \in \{1,..,K\}} s_k^{i,j}(t))$ and the worst return to compute the IQM of SEARL-min $(\min_{k \in \{1,..,K\}} s_k^{i,j}(t))$.

Table 1: Summary of all fixed hyperparameters used for the Lunar Lander experiments (see Section 5.1).

| Shared across all algorithms | |
|---|---|
| Discount factor $\gamma$ | 0.99 |
| Horizon | 1000 |
| Initial replay buffer size | 1000 |
| Replay buffer size | $10^4$ |
| Batch Size | 32 |
| Target update frequency $T$ | 200 |
| Training frequency $G$ | 1 |
| Activation function | ReLU |
| Learning rate | $3 \times 10^{-4}$ |
| Optimizer | Adam |
| Loss | L2 |
| Starting $\epsilon$ | 1 |
| Ending $\epsilon$ | 0.01 |
| Linear decay duration of $\epsilon$ | 1000 |
| AdaDQN | |
| Starting $\epsilon_b$ | 1 |
| Ending $\epsilon_b$ | 0.01 |
| Linear decay duration of $\epsilon_b$ | Until end |

Table 2: Summary of all fixed hyperparameters for the MuJoCo experiments (see Section 5.2).

| Shared across all algorithms | |
|---|---|
| Discount factor $\gamma$ | 0.99 |
| Horizon | 1000 |
| Initial replay buffer size | 5000 |
| Replay buffer size | $10^6$ |
| Batch Size | 256 |
| Target update rate $\tau$ | 0.005 |
| UTD | 1 |
| Actor's activation function | ReLU |
| Actor's learning rate | 0.001 |
| Actor's optimizer | Adam |
| Actor's architecture | $256, 256$ |
| Policy delay | 1 |
| Vanilla SAC | |
| Critic's activation function | ReLU |
| Critic's learning rate | 0.001 |
| Critic's optimizer | Adam |
| Critic's architecture | $256, 256$ |
| AdaSAC | |
| Starting $\epsilon_b$ | 1 |
| Ending $\epsilon_b$ | 0.01 |
| Linear decay duration of $\epsilon_b$ | Until end |

Table 3: Summary of all fixed hyperparameters used for the Atari experiments with a *finite* hyperparameter space (see Section 5.3).

| Shared across all algorithms | |
|---|---|
| Discount factor $\gamma$ | 0.99 |
| Horizon | 27000 |
| Full action space | No |
| Reward clipping | clip$(-1, 1)$ |
| Initial replay buffer size | 20000 |
| Replay buffer size | $10^6$ |
| Batch Size | 32 |
| Target update frequency $T$ | 8000 |
| Training frequency $G$ | 4 |
| Learning rate | $5 \times 10^{-5}$ |
| Optimizer | Adam |
| Loss | L2 |
| Starting $\epsilon$ | 1 |
| Ending $\epsilon$ | 0.01 |
| Linear decay duration of $\epsilon$ | 250000 |
| **Vanilla DQN** | |
| Activation function | ReLU |
| $\epsilon_{\text{Adam}}$ | $1.5 \times 10^{-4}$ |
| CNN n layers | 3 |
| CNN channels | $[32, 64, 64]$ |
| CNN kernel size | $[8, 4, 3]$ |
| CNN stride | $[4, 2, 1]$ |
| MLP n layers | 1 |
| MLP n neurons | 512 |
| **AdaDQN** | |
| Starting $\epsilon_b$ | 1 |
| Ending $\epsilon_b$ | 0.01 |
| Linear decay duration of $\epsilon_b$ | Until end |

Table 4: Summary of all hyperparameters used for the Atari experiments with an *infinite* hyperparameter space (see Section 5.4).

| Shared across all algorithms | |
|---|---|
| Discount factor $\gamma$ | 0.99 |
| Horizon | 27000 |
| Full action space | No |
| Reward clipping | clip$(-1, 1)$ |
| Initial replay buffer size | 20000 |
| Replay buffer size | $10^6$ |
| Batch Size | 32 |
| Training frequency $G$ | 4 |
| Starting $\epsilon$ | 1 |
| Ending $\epsilon$ | 0.01 |
| Linear decay duration of $\epsilon$ | 250000 |
| **Hyperparameter space** | |
| Activation function | {CELU, ELU, GELU, Hard Sigmoid, Hard SiLU, Hard tanh, Leaky ReLU, Log-Sigmoid, Log-Softmax, ReLU, SELU, Sigmoid, SiLU, Soft-sign, Softmax, Softplus, Standardize, Tanh} |
| Learning rate (in log space) | $[10^{-6}, 10^{-3}]$ |
| Optimizer | {AdaBelief, AdaDelta, AdaGrad, AdaFactor, Adam, Adamax, AdamaxW, AdamW, AMSGrad, Fromage, Lamb, Lars, Lion, Nadam, NadamW, Noisy SGD, Novograd, Optimistic GD, RAdam, RMSProp, RProp, SGD, SM3, Yogi} |
| Loss | {Huber, L1, L2, Log cosh} |
| CNN n layers | $[\![1, 3]\!]$ |
| CNN n channels | $[\![16, 64]\!]$ |
| CNN kernel size | $[\![2, 8]\!]$ |
| CNN stride | $[\![2, 5]\!]$ |
| MLP n layers | $[\![0, 2]\!]$ |
| MLP n neurons | $[\![25, 512]\!]$ |
| **Different Settings** | |
| High target freq | $T = 8000$ |
| Low target freq | $T = 4000$ |
| High HP freq | $M = 80000$ |
| Low HP freq | $M = 40000$ |

## C TRAINING TIME AND MEMORY REQUIREMENTS

We address time complexity by reporting the performances obtained in Section 5.2 and 5.4 with the clock time as the horizontal axis in Figure 10. On the left, AdaSAC still outperforms grid search but is slower than random search as AdaQN carries out several networks in parallel. We stress that this experiment is designed to demonstrate the capabilities of AdaSAC when a high number of hyperparameters is considered. When the number of network decreases to 5, AdaQN becomes more competitive against random search as Figure 10 (right) testifies. In this case, AdaDQN outperforms all baselines in terms of training time and sample efficiency. We point out that Meta-gradient DQN, evaluated in Section 5.4, is 4% slower to run than AdaDQN. Finally, we recall that, in this work, the main focus is on sample-efficiency, as it is the predominant bottleneck in practical scenarios.

Concerning the memory requirements, AdaSAC uses an additional 300Mb from the vRAM for the MuJoCo experiments presented in Section 5.2 compared to SAC. AdaDQN uses 200Mb more from the vRAM than random search for the Atari experiments presented in Section 5.4. We argue that those increases remain small compared to the benefit brought by AdaQN.

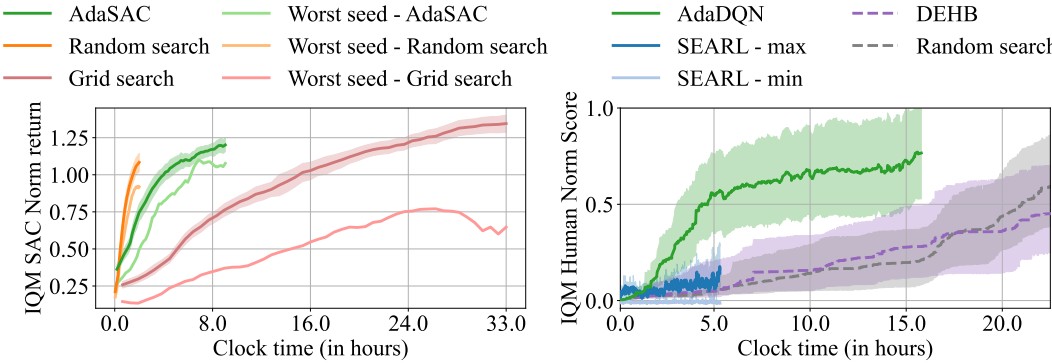

Figure 10: Performances observed in Section 5.2 (left) and Section 5.4 (right) according to the training time. When evaluated with a high number of hyperparameters to consider in parallel, AdaQN suffers from a slower training time but is still faster than grid search. When the number of hyperparameters is reasonable ($\approx 5$), AdaQN is competitive against all baselines.

## D ADDITIONAL EXPERIMENTS

### D.1 ON-THE-FLY HYPERPARAMETER SELECTION ON MUJOCO

- - - SAC learning rate: 0.001, optimizer: Adam, architecture: 512, 512, activation fn: ReLU
- - - SAC learning rate: 0.0005, optimizer: Adam, architecture: 256, 256, activation fn: ReLU
- - - SAC learning rate: 0.001, optimizer: Adam, architecture: 512, 512, activation fn: Sigmoid
- - - SAC learning rate: 0.0005, optimizer: Adam, architecture: 512, 512, activation fn: ReLU
- - - SAC learning rate: 0.001, optimizer: Adam, architecture: 256, 256, activation fn: Sigmoid
- - - SAC learning rate: 0.0005, optimizer: Adam, architecture: 512, 512, activation fn: Sigmoid
- - - SAC learning rate: 0.0005, optimizer: RMSProp, architecture: 256, 256, activation fn: ReLU
- - - SAC learning rate: 0.0005, optimizer: RMSProp, architecture: 512, 512, activation fn: ReLU
- - - SAC learning rate: 0.001, optimizer: RMSProp, architecture: 512, 512, activation fn: Sigmoid
- - - SAC learning rate: 0.0005, optimizer: Adam, architecture: 256, 256, activation fn: Sigmoid
- - - SAC learning rate: 0.001, optimizer: RMSProp, architecture: 256, 256, activation fn: ReLU
- - - vanilla SAC learning rate: 0.001, optimizer: Adam, architecture: 256, 256, activation fn: ReLU
- - - SAC learning rate: 0.001, optimizer: RMSProp, architecture: 256, 256, activation fn: Sigmoid
- - - SAC learning rate: 0.001, optimizer: RMSProp, architecture: 512, 512, activation fn: ReLU
- - - SAC learning rate: 0.0005, optimizer: RMSProp, architecture: 512, 512, activation fn: Sigmoid
- - - SAC learning rate: 0.0005, optimizer: RMSProp, architecture: 256, 256, activation fn: Sigmoid

Figure 11: Exhaustive legend of Figure 4 showing the ranking by AUC of the 16 considered sets of hyperparameters. Interestingly, the ranking does not follow a predictable order, as there is no clear ordering between each hyperparameter. Importantly, the ranking changes between MuJoCo environments, justifying the usage of AdaSAC, which adapts the hyperparameter schedule for *each* environment and *each* seed.

## D.2 ON-THE-FLY ARCHITECTURE SELECTION ON MUJOCO

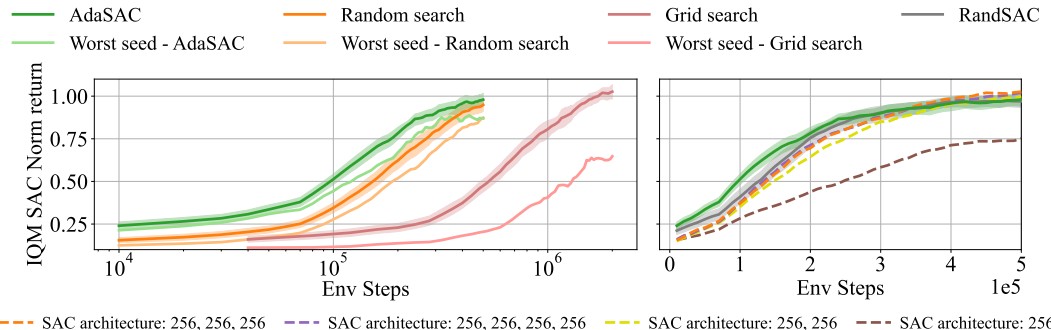

Figure 12: On-the-fly architecture selection on **MuJoCo**. All architectures contain hidden layers with 256 neurons. The architectures are indicated in the legend. **Left:** AdaSAC is more sample-efficient than grid search. **Right:** AdaSAC yields similar performances as the best-performing architectures. RandSAC and AdaSAC are on par.

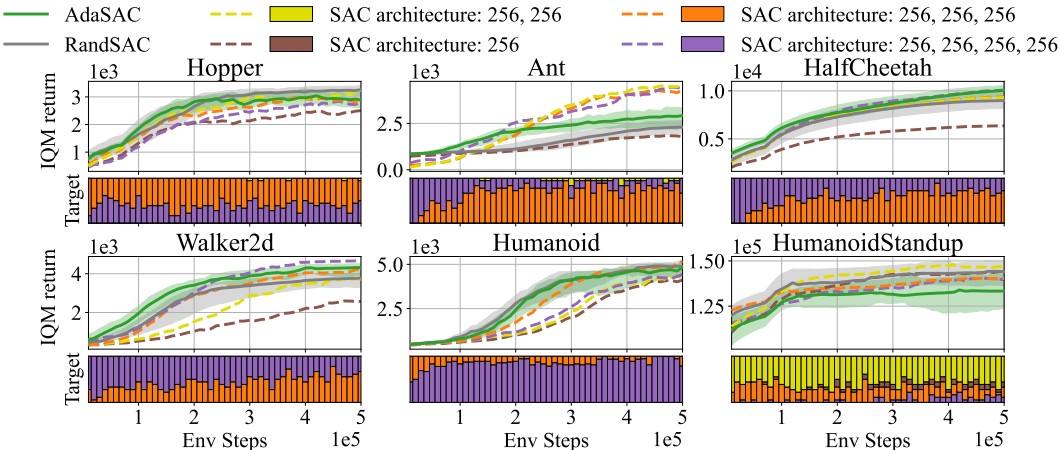

Figure 13: Per environment return of AdaSAC and RandSAC when 4 different architectures are given as input. Below each performance plot, a bar plot presents the distribution of the hyperparameters selected for computing the target across all seeds. AdaSAC effectively ignores the worst-performing architecture (256) while actively selecting the best-performing architecture. Interestingly, in *HumanoidStandup*, despite not performing well, AdaSAC selects the architecture 256, 256, which is the best-performing-architecture.

### D.3    ON-THE-FLY LEARNING RATE SELECTION ON MUJOCO

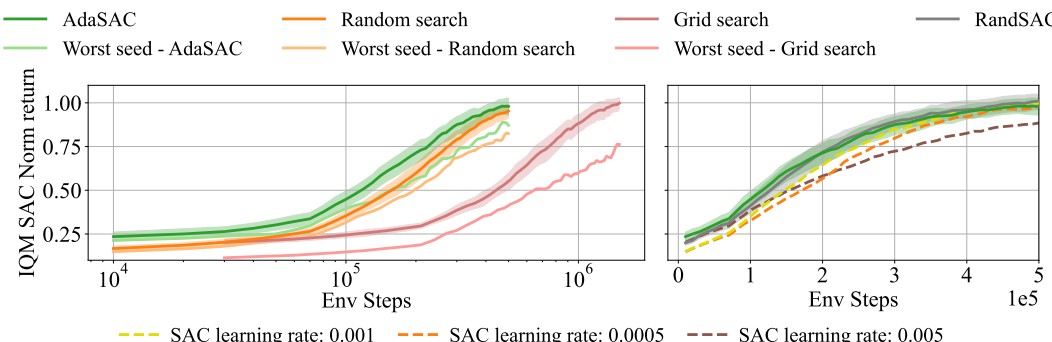

Figure 14: On-the-fly learning rate selection on **MuJoCo**. The space of hyperparameters is composed of 3 different learning rates. **Left:** AdaSAC is more sample-efficient than random search and grid search. **Right:** AdaSAC yields performances slightly above the best-performing learning rate. RandSAC and AdaSAC are on par.

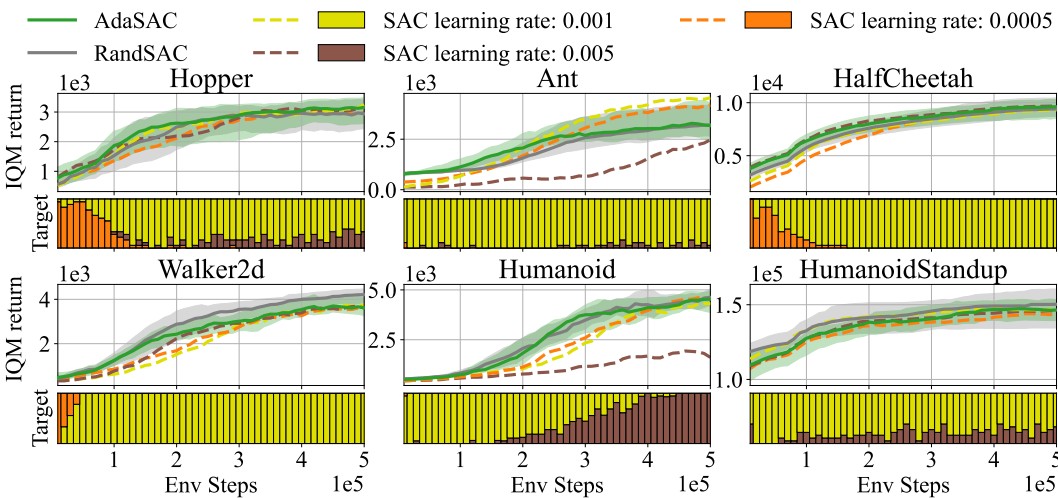

Figure 15: Per environment return of AdaSAC and RandSAC when 3 different learning rates are given as input. Below each performance plot, a bar plot presents the distribution of the hyperparameters selected for computing the target across all seeds. AdaSAC creates a custom learning rate schedule for each environment and each seed. Interestingly, this schedule seems to choose a low learning rate at the beginning of the training before increasing it later during the training.

## D.4 ON-THE-FLY OPTIMIZER SELECTION ON MUJOCO

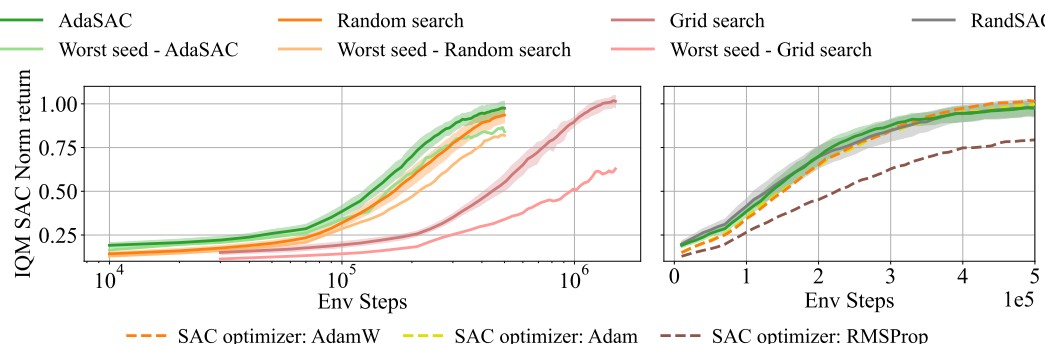

Figure 16: On-the-fly optimizer selection on **MuJoCo**. The space of hyperparameters is composed of 3 different optimizers. **Left:** AdaSAC is more sample-efficient than random search and grid search. **Right:** AdaSAC yields similar performances as the best-performing optimizer (AdamW, Loshchilov & Hutter (2018)). RandSAC and AdaSAC are on par.

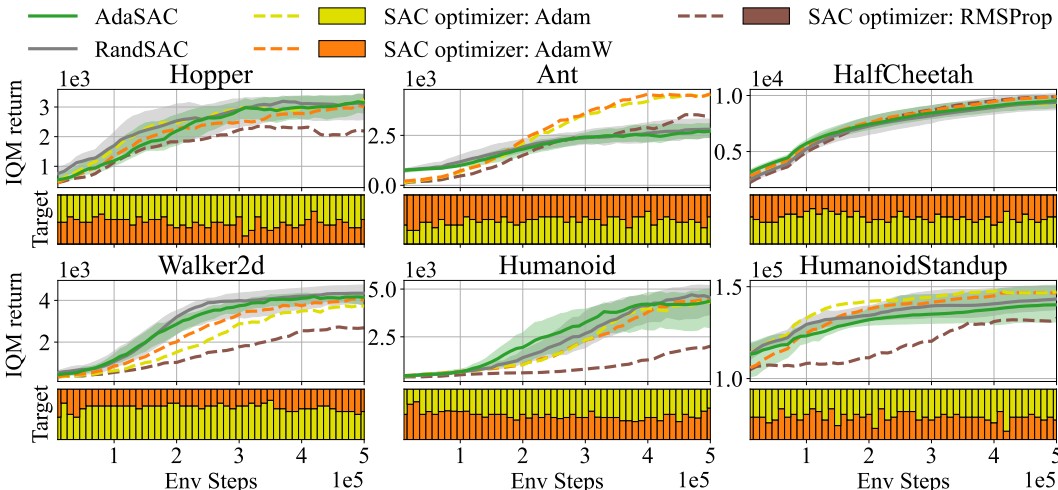

Figure 17: Per environment return of AdaSAC and RandSAC when 3 different optimizers are given as input. Below each performance plot, a bar plot presents the distribution of the hyperparameters selected for computing the target across all seeds. AdaSAC effectively ignores RMSProp, which performs poorly in most environments.

## D.5    ON-THE-FLY ACTIVATION FUNCTION SELECTION ON MUJOCO

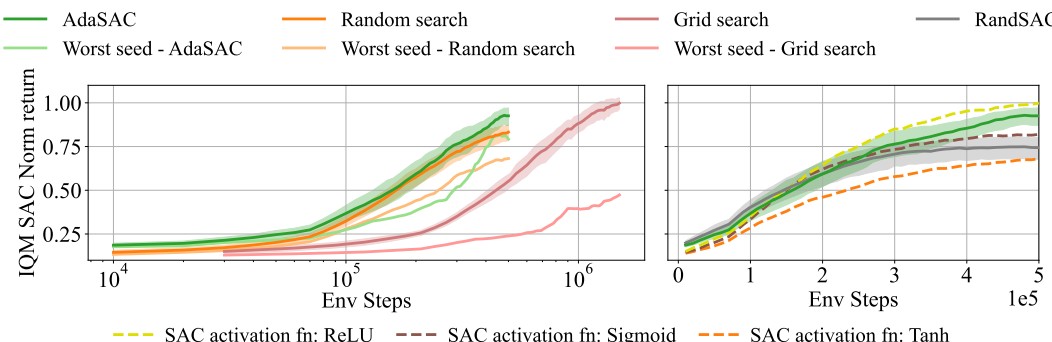

Figure 18: On-the-fly activation function selection on **MuJoCo**. The space of hyperparameters is composed of 3 different activation functions. **Left:** AdaSAC is more sample-efficient than random search and grid search. **Right:** AdaSAC performs slightly below the best-performing activation function. In this setting, RandSAC suffers from the fact that a network with Tanh activation functions is selected as the target network one-third of the time.

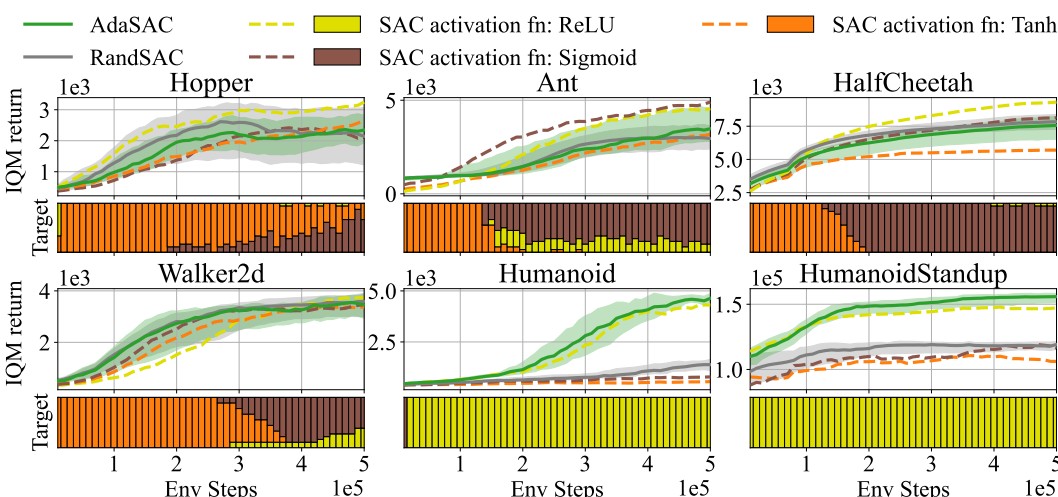

Figure 19: Per environment return of AdaSAC and RandSAC when 3 different activation functions are given as input. Below each performance plot, a bar plot presents the distribution of the hyperparameters selected for computing the target across all seeds. Remarkably, AdaSAC selects the Sigmoid activation function in environments where this activation function seems beneficial (*Ant*, *HalfCheetah*, and *Walker2d*), while it does not select the Sigmoid activation function in environments where the individual run performs poorly (*Humanoid* and *HumanoidStandup*).

## D.6  ON-THE-FLY HYPERPARAMETER SELECTION ON ATARI

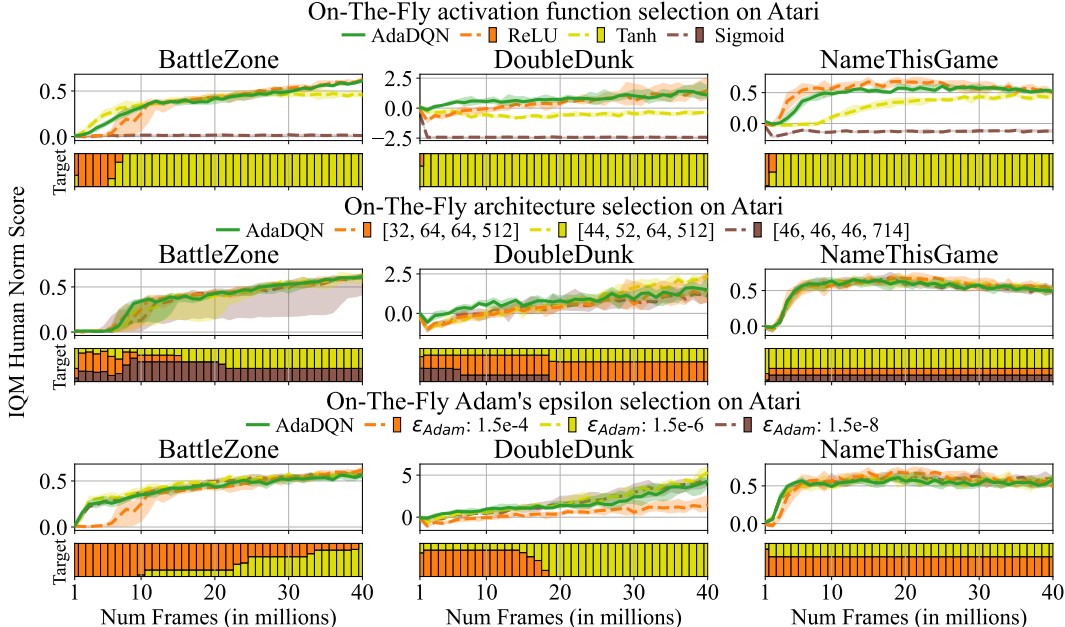

Figure 20: AdaDQN performs similarly to the best individual hyperparameters when selecting from 3 activation functions (left), 3 architectures (middle), or 3 different values of Adam's epsilon parameter (right). Below each performance plot, a bar plot presents the distribution of the hyperparameters selected for computing the target.

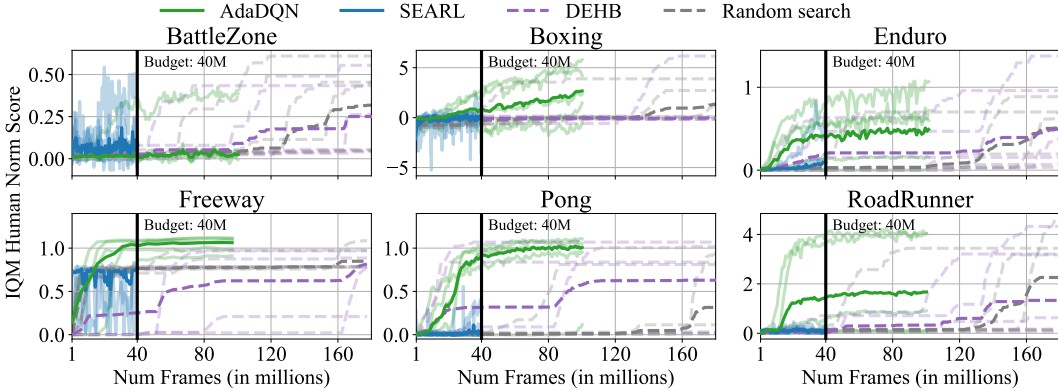

Figure 21: Per game performances of AdaDQN and the baselines when the space of hyperparameters is infinite. We believe that the wide range of returns is due to the fact that the hyperparameter space is large. Nevertheless, AdaDQN reaches higher IQM returns than SEARL on all games except on *BattleZone*, where only one seed performs well.

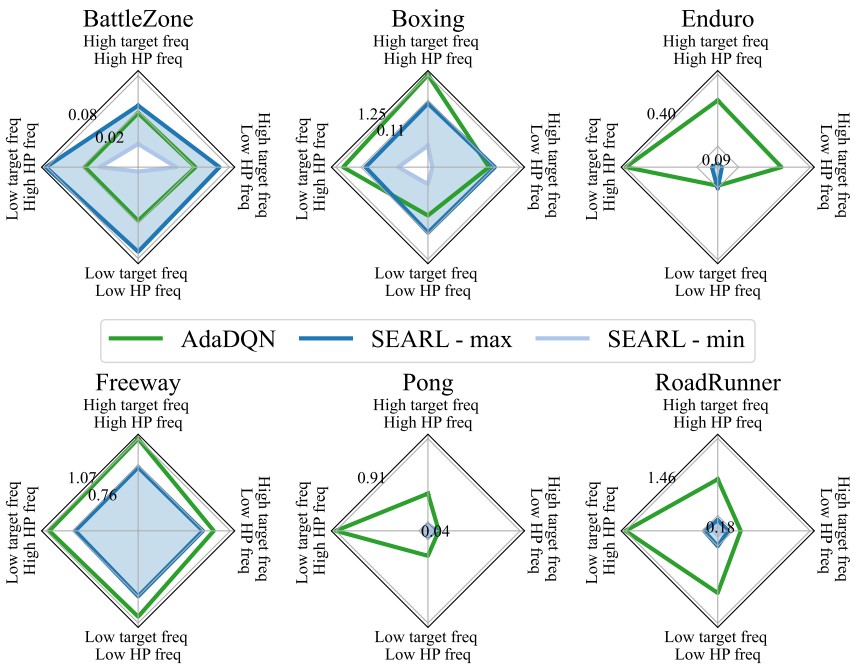

Figure 22: Per game performances of AdaDQN and SEARL at $40M$ frames for different algorithm's hyperparameter values. As each online network is evaluated in SEARL, we report the maximum and minimum observed return (SEARL - min and SEARL - max). Remarkably, AdaDQN's performances are always above the minimum performances of SEARL. In 4 out of the 6 considered games, AdaDQN outperforms SEARL maximum return on all considered settings. High target freq corresponds to $T = 8000$ and low target freq corresponds to $T = 4000$. HP freq is noted $M$ in Figure 9, the high value is $M = 80000$ and the low value is $M = 40000$.

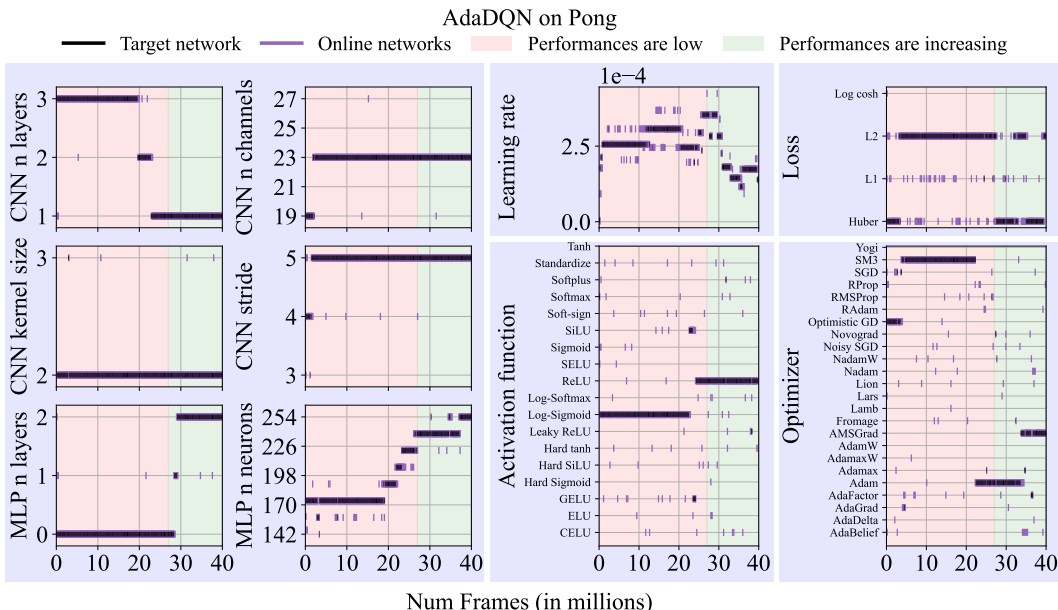

Figure 23: Evolution of AdaDQN's hyperparameters along the training for one seed on *Pong*. While the online networks explore a vast portion of the hyperparameter space, the target network focuses on a few values.

