# OpenReview forum: "Adaptive $Q$-Network: On-the-fly Target Selection for Deep Reinforcement Learning"
_ICLR.cc/2025/Conference — ICLR 2025 Poster_

### Official Review · Reviewer_PnAT · 2024-11-02

**Soundness:** 3
**Presentation:** 3
**Contribution:** 3
**Rating:** 6
**Confidence:** 3

**Summary:**

This paper addresses the sensitivity of hyperparameters in deep Reinforcement Learning (RL) and proposes a method to speed up hyperparameter selection compared to previous AutoRL approaches. The authors introduce the Adaptive Q-Network (AdaQN), which learns several Q-functions, each trained with different hyperparameters. These Q-functions are updated online using the one with the smallest approximation error as a shared target. Comprehensive experiments on MuJoCo and Atari environments demonstrate the effectiveness of AdaQN.

**Strengths:**

This paper addresses the crucial issue of hyperparameter selection in deep RL and provides an effective solution. The proposed method is well written and organized. Comprehensive experiments validate the effectiveness of the method.

**Weaknesses:**

Theorem 4.1 explains why the empirical Bellman operator in Eq (3) derived from experience replay is unbiased to the true Bellman operator in Eq (2). However, it lacks a theoretical analysis on why the selection scheme from Eq (2) leads to convergence rather than local minima. A formal theoretical analysis would strengthen the paper by explaining the convergence properties of the proposed method.

**Questions:**

Given the motivation to "speed up hyperparameters selection," how can the selected hyperparameters by AdaQN be used effectively? Can the selected hyperparameters by AdaQN be used to train a better DQN? It seems the improvements by AdaQN are due to the use of different Q-functions, similar to ensemble methods, with the difference being the use of various hyperparameters instead of just different initializations. Could the paper be motivated by performance improvement using different hyperparameters rather than speeding up hyperparameter selection?

---

> ### Author Response · Authors · 2024-11-20
>
> We thank the reviewer for the insightful comments and the time spent reviewing the paper.
>
> > Weakness 1: [...] it lacks a theoretical analysis on why the selection scheme from Eq (2) leads to convergence rather than local minima. A formal theoretical analysis would strengthen the paper by explaining the convergence properties of the proposed method.
>
> We thank the Reviewer for their suggestion. In Appendix A.1 of the updated submission, we now prove that AdaQN is guaranteed to converge to the optimal action-value function in the tabular setting with probability $1$.
>
> *Sketch of the proof:* we demonstrate AdaQN's convergence by adapting the proof of convergence of the Generalized Q-learning framework presented in [1]. For that, we first explain which modifications to the Generalized Q-learning framework are needed to englobe AdaQN. Then, we prove that the assumptions required for convergence still hold with the presented modification.
>
> [1] Q. Lan, et al. Maxmin q-learning: Controlling the estimation bias of q-learning. ICLR, 2020.
>
> > Question 1: Given the motivation to "speed up hyperparameters selection," how can the selected hyperparameters by AdaQN be used effectively?
>
> We stress that the purpose of AdaQN is not to pre-select some hyperparameters that could be used to train a well-performing agent from scratch. In this work, we argue and demonstrate that training a method with fixed hyperparameters is suboptimal compared to selecting (and changing) them **dynamically**, thus better coping with the high non-stationarity of the RL training process.
>
> If more environment interactions are available, we recommend continuing the training with AdaQN. Especially when the search space is infinite (Section $5.4$), AdaQN will continue to explore the space of hyperparameters while focusing on the hyperparameters that fit the target well. Thanks to the offline selection mechanism, no environment iteration will be wasted on exploration. Additionally, the population size can be reduced if the computation power is a limiting factor.
>
>
> > Question 2: Can the selected hyperparameters by AdaQN be used to train a better DQN? It seems the improvements by AdaQN are due to the use of different Q-functions, similar to ensemble methods, with the difference being the use of various hyperparameters instead of just different initializations. Could the paper be motivated by performance improvement using different hyperparameters rather than speeding up hyperparameter selection?
>
> As mentioned in the answer to Question $1$, we point out that the main goal of this work is not to speed up hyperparameter selection. The main objective is to improve performances and remove the need for expert knowledge on hyperparameters. Our approach selects the hyperparameters dynamically during training to cope with the non-stationarity of the RL process, as we demonstrate that keeping fixed hyperparameters is suboptimal. As pointed out by the Reviewer and mentioned in the last paragraph before Section 4.1, "By cleverly selecting the next target network from a set of diverse online networks, AdaQN has a higher chance of overcoming the typical challenges of optimization".
>
> The results of Figure $6$ is an example where AdaQN is used to train a better DQN. Indeed, AdaDQN consistently outperforms the orange curve which corresponds to the hyperparameters of vanilla DQN. In those $3$ configurations, adding diversity allows AdaDQN to outperform vanilla DQN.

---

> > ### Comment · Reviewer_PnAT · 2024-11-23
> >
> > I appreciate the authors for providing  a thorough explanation that resolves my concerns.
> > I have raised my score accordingly.

---

> > > ### Author Response · Authors · 2024-11-23
> > >
> > > We are glad to see our rebuttal resolved the Reviewer's concerns. We thank the Reviewer for raising their score.

---

### Official Review · Reviewer_Kj2W · 2024-11-05

**Soundness:** 3
**Presentation:** 3
**Contribution:** 3
**Rating:** 6
**Confidence:** 3

**Summary:**

This work proposes a new AutoRL algorithms that constructs an ensemble of Q-networks, each learned with different hyperaparameters (e.g., different learning rates). The main novelty is that during learning, there is an interaction between the different Q-networks; namely, the best performing Q-network is used for the target computation for all networks. This means that the individual performances of the networks will be different, potentially better than the performance obtained by just learning by scratch with a chosen hyperparameter. Moreover, the best performing hyperparameters may also change as learning progresses, highlighting an attractive property of the proposal. Action selection is also performed by picking the best performing Q-network from the previous transition, together with an $\epsilon$ random selection of a Q-network from the ensemble. Experiments are performed on DQN lunar lander as well as SAC on MuJoCo tasks. They compare against other AutoRL algorithms like SEARL and DEHB, and obtain superior performance. The work also includes ablation studies on the components of the algorithm, e.g., the epsilon parameter, and the Q-network selection mechanism. Moreover, they provide other evidence, such as directly plotting which networks were selected. The final performances on the tasks seem reasonable, and the learning speed including hyperparameter selection is good compared to the baselines.

**Strengths:**

- The main idea is clear, theoretically justified and seems like a good idea.
- The performance improves significantly.
- The experiments are thorough, and look at several metrics, include ablations, etc.
- The exposition is good.

**Weaknesses:**

- The evaluation protocol was only briefly discussed, and was not completely clear to me. Particularly, how the baselines results are plotted could have been explained more prominently. I believe these plots include the hyperparameter selection cost on x-axis, but this could be clearer. Moreover, for reference, it would be good to know what the performance would be when running from scratch with the best performing hyperparameters.

In particular, what does this mean:
> Additionally, we focus on sample efficiency; thus, for reporting the grid search results, we multiply the number of interactions of the best achieved performance by the number of individual trials, as advocated by Franke et al. (2021). We also report the average performance over the individual runs as the performance that a random search would obtain in expectation.

This bit seems very important for interpreting the results, but it's only briefly mentioned. I would suggest explaining this with equation notation to make it precise. Moreover, perhaps it would be useful to explain the hyperparameter tuning task setting, what the objective and metrics are in the background. Moreover, the axis labels such as "Env steps" can be confusing, because the same notation is used in standard RL where one just runs the best performing hyperparameters without considering cost of searching the hyperparameters. These points would be good to clarify.

- Around the end of the paper, there was a part about also selecting the $\gamma$ discount parameter, but this seemed unsound to me. The discount parameter affects the Q-function, so different Q-functions would necessarily require different targets, and I don’t think they could be shared. For example setting $\gamma=0$ would probably give good small prediction errors, but would not actually perform well.

**Questions:**

Can you give more explanation on the evaluation protocol?

What is the performance when running the optimal hyperparameters from the start?

Can you clarify the parts that I listed in the weaknesses, including the discussion on the discount factor?

---

> ### Author Response · Authors · 2024-11-20
>
> We thank the Reviewer for the thorough comments and questions.
>
> ## Weakness 1/Question 1: Evaluation protocol
> > It would be good to know what the performance would be when running from scratch with the best performing hyperparameters.
> > I would suggest explaining this with equation notation to make it precise.
> > Moreover, perhaps it would be useful to explain the hyperparameter tuning task setting, what the objective and metrics are in the background.
>
> Thank you for pointing this out. We incorporated the following in Appendix $B.1$ in the updated submission. We now detail the evaluation protocol for the considered methods. The reported performance of the individual runs of DQN and SAC as well as AdaDQN, AdaDQN with $\epsilon_b = 0$, RandDQN, AdaDQN-max, AdaSAC, RandSAC, and Meta-Gradient DQN are the raw performance obtained during training aggregated over seeds and environments. The reported performances of the other methods require some post-processing steps to account for the way they operate:
> - **Random search:** we report the empirical average over all the possible individual runs of the hyperparameter search space in Figure $4$. This is possible because the search space is finite. In Figure $7$ (left), we had to adapt this protocol as the search space is infinite. This is why we computed the best performance observed before a given timestep as the current performance before computing the IQM over the seeds.
> - **DEHB:** the same protocol as a random search was used. The average return over the last epoch is given to the hyperparameter optimizer as a metric to be maximized.
> - **Grid search:** we first compute the IQM performance of each individual run aggregated over seeds and tasks. Then, we compute the IQM of the best individual run and scale the resulting IQM horizontally to the right by the number of individual runs to account for the environment steps taken by all individual runs. If we note $p_k(t)$ the IQM obtained by a hyperparameter $k$ after $t$ environment steps. The IQM of grid search is reported as $p_{\text{grid search}}(t \times N) = \max_{k \in \{1, .., K\}} p_k(t)$ where $K$ is the number of individual runs used in the grid search.
> - **SEARL:** we recall that SEARL has the specificity of periodically evaluating every $Q$-network of its population. Each time every $Q$-network is evaluated, we use the best return to compute the IQM of SEARL-max and the worst return to compute the IQM of SEARL-min.
>
> Details on the hyperparameter spaces and the other hyperparameters are available in Tables 1, 2, 3, and 4 of Appendix $B.1$.
>
> > Moreover, the axis labels such as "Env steps" can be confusing, because the same notation is used in standard RL where one just runs the best performing hyperparameters without considering cost of searching the hyperparameters.
>
> We agree that some works do not share the number of environment steps used for tuning the method's hyperparameters. We argue that this is a bad practice as it does not reflect the dependency of the proposed algorithm w.r.t its hyperparameters. It can also lead to wrong conclusions when less effort has been spent on tuning the baseline's hyperparameters. Therefore, we include all the environment steps used for achieving the reported performances, as advocated by [1].
>
> [1] J. K.H. Franke, G. Koehler, A. Biedenkapp, and F. Hutter. Sample-efficient automated deep reinforcement learning. ICLR, 2021.
>
>
> ## Weakness 2/Question3: Selecting from different discount factor
> > Around the end of the paper, there was a part about also selecting the discount parameter, but this seemed unsound to me. The discount parameter affects the Q-function, so different Q-functions would necessarily require different targets, and I don’t think they could be shared. For example setting $\gamma = 0$ would probably give good small prediction errors, but would not actually perform well.
>
> Thank you for raising this point. While each $Q$-function is learned w.r.t. a different discount factor, they are all compared against the same discount factor $\gamma = 0.99$. This prevents situations where a smaller discount factor would be favored as suggested by the Reviewer. This mechanism is close to the way Meta-Gradient DQN works. Indeed, Meta-Gradient DQN uses a cross-validated discount factor for computing the meta-objective. As indicated in the last paragraph of Section $5.4$, we choose the same value for the cross-validated discount factor and the shared discount factor so that the algorithms can be compared fairly.

---

> > ### Author Response · Authors · 2024-11-20
> >
> > ## Question 2: What is the performance when running the optimal hyperparameters from the start?
> > We stress that the purpose of AdaQN is not to pre-select some hyperparameters that could be used to train a well-performing agent from scratch with fixed hyperparameters. In this work, we argue and demonstrate that training a method with fixed hyperparameters is suboptimal compared to changing them dynamically as the RL training process is non-stationary w.r.t to the optimal hyperparameter. Indeed, in Section $4$, we argue that having dynamic hyperparameters lowers the risk of facing local minima or diverging behaviors. Then, we show in Section $5$ that AdaQN performs better or on par with the best static hyperparameters.
> >
> > Nonetheless, we report in Figure $5$ the hyperparameters that were selected for computing the target during training. It is represented as a bar plot because different hyperparameters were selected across the seeds. Each color represents a different hyperparameter. Greener colors are used for hyperparameters that perform better when trained individually. The performance of each individual training is reported in the plot above each bar plot. This means that bar plots that are mostly green indicate the AdaSAC selects hyperparameters that perform well when trained individually. This is the case for *Hopper*, *HalfCheetah*, *Humanoid* and *HumanoidStandup*. Interestingly, in *Walker2d*, while AdaSAC performs similarly to the best hyperparameter, AdaSAC mainly selects hyperparameters that perform poorly when trained individually. This highlights the fact that changing hyperparameters dynamically during training can lead to high performances even when "bad" hyperparameters are selected. This is why, if more environment interactions are available, we recommend continuing the training instead of training an agent from scratch with fixed hyperparameters.

---

> > > ### Comment · Reviewer_Kj2W · 2024-11-24
> > >
> > > Thank you for the response.
> > >
> > > Mostly, my questions were addressed, in particular, I now understand how the $\gamma$ experiment was performed.
> > >
> > > Regarding the evaluation protocols, I think it could still be explained more clearly with equation notation, writing out precisely what was done. I also still think that the "Env steps" notation can be confusing, so I would suggest thinking about whether it could be revised somehow. I do agree that looking at all of the steps used for hyperparameter selection is meaningful, but I just think the notation is easily confused with standard RL. Maybe something like "Hyperopt Env Steps" or something else along those lines would message to the reader that the plots are not in the standard RL setting.
> > >
> > > I encourage the authors to further revise the notation and explanations. Perhaps explaining the autoRL framework with mathematical notation in the preliminaries or background would also be useful to set the stage and clarify precisely what is being aimed to do in the paper.
> > >
> > > I think this is a high quality paper, but I also think it's not quite enough for a higher score, so I will be sticking to my current score.

---

> ### Author Response · Authors · 2024-11-25
>
> We thank the Reviewer for the time spent on our rebuttal. We are glad to see that our rebuttal addressed their questions.
>
> > Regarding the evaluation protocols, I think it could still be explained more clearly with equation notation, writing out precisely what was done.
>
> We thank the Reviewer for their suggestion. We now further refined Appendix $A.1$ in the updated submission, where the equation used for computing the reported IQMs of each method is written down.
>
> > I also still think that the "Env steps" notation can be confusing, so I would suggest thinking about whether it could be revised somehow.
>
> We understand the Reviewer's concern. However, we would like to keep "Env steps" as a label because we believe it is an accurate description of the quantity it represents. Nonetheless, we now emphasized that "the environment steps used for tuning the hyperparameters are also accounted for when reporting the results" in Line $267$ of Section $5$ in the updated submission.
>
> > Explaining the autoRL framework with mathematical notation in the preliminaries or background would also be useful to set the stage and clarify precisely what is being aimed to do in the paper.
>
> Thank you. We now added a sentence in the first paragraph of the introduction that clearly states the scope of the paper. We believe that this is enough to properly understand the AutoRL framework under which our approach is developed.

---

> > ### Comment · Reviewer_Kj2W · 2024-11-25
> >
> > I still don't quite understand the setting.
> >
> > For example, you write:
> > >Random search: we report the empirical average over all the possible individual runs
> > of the hyperparameter search space in Figure 4. The IQM of random search is reported
> > as $p_{random search} = \frac{1}{K}\sum p_k(t)$.
> >
> > But, the goal of hyperparameter search is to find the best performing hyperparameter. I would have expected the protocol to compute the expected maximum of the random search after $t$ steps. This seems to however compute the average irrespective of the number of steps, so it would never converge to the performance of the maximum hyperparameter.
> >
> > Regarding the explanation of the setting, what I had in mind was to write out in equations that you are running a set of trials, where each trial has some number of steps $T$. Each trial can have either different hyperparameters or the hyperparameters may also be tuned adaptively. You are trying to find the best performing policy compared to the total number of environment steps considered, etc... This kind of explanation would clarify what the exact setting is. Indeed it is possible to infer these points from the paper at the moment, but I think it would improve the clarity to have this written out precisely.
> >
> > Currently, my main concern is the question about the protocol that I just asked.

---

> ### Author Response · Authors · 2024-11-26
>
> Thank you for your detailed comments.
>
> > a. I would have expected the protocol to compute the expected maximum of the random search after $t$ steps.
>
> We stress that this protocol is only applied in Figure $4$ in which the search space is composed of $16$ hyperparameters. We first computed all possible individual runs (Figure $4$ (right)) and, in Figure $4$ (left), we derive:
>
> - **Grid search** as the IQM of the best individual run horizontally scaled to the right by the number of individual runs to account for the environment steps taken by all individual runs. The IQM of grid search is reported as $p_{\text{grid search}}(t \times K) = \max_{k \in \{1, .., K\}} p_k(t)$.
>
> - **Random search** as the empirical average over all individual runs. The IQM of random search is reported as $p_{\text{random search}}(t) = \frac{1}{K} \sum_{k \in \{1, .., K\}} p_k(t)$. This corresponds to the expected performance of what a random search would look like if only one individual trial were used. We chose to add this metric to show that AdaSAC is a better algorithm than simply picking a random hyperparameter.
>
> This clarification is now part of Appendix $B.2$ in the updated submission.
>
> > b. This seems to however compute the average irrespective of the number of steps
>
> We clarify that the number of steps is taken as input such that $p_{\text{random search}}(t) = \frac{1}{K} \sum_k p_k(t)$. Note that we forgot the argument $t$ for $p_{\text{random search}}$ in the previous revision; we have now added it.
>
> > c. it would never converge to the performance of the maximum hyperparameter.
>
> As mentioned in the answer to Point a., the curve of random search is only shown for the duration of one individual run because the purpose is simply to show that AdaSAC is a better algorithm than simply picking a random hyperparameter.
>
> Continuing the curve for random search would lead to the performance of the maximum hyperparameter. However, computing the expected performance of random search beyond the first individual run is computationally demanding as $16! \approx 2.0 \times 10^{13}$ combinations have to be taken into account. Moreover, we argue that the most relevant part of this curve is the first individual run as AdaSAC only uses this amount of environment interactions. Therefore, we decided not to compute the curve for random search after the first individual run.
>
> > Regarding the explanation of the setting, what I had in mind was to write out in equations that you are running a set of trials, where each trial has some number of steps $T$. Each trial can have either different hyperparameters or the hyperparameters may also be tuned adaptively. You are trying to find the best performing policy compared to the total number of environment steps considered, etc... This kind of explanation would clarify what the exact setting is.
>
> We thank the Reviewer for the suggestion. In the updated submission, we clarified the AutoRL setting in which our work stands by formulating the following optimization problem that our approach aims to solve:
>
> $\max_{k} J(\theta_t^k) s.t. \forall k \in \{1, .., K\}, \theta_t^k \in \text{arg} \min_{\theta} f(\theta; \zeta_t^k),$
>
> where $t$ is the current timestep, $K$ is the population size, $\theta_t^k$ is the $k^{\text{th}}$ element of the population, $J$ is the cumulative discounted return, and $f$ represents the objective function of the learning process.
>
> Indeed, as explained in the last paragraph of Section $4$, at each timestep $t$, AdaQN aims at using the best agent of the population. Since the identity of the best agent is not known without evaluating each agent explicitly (as done for instance by SEARL), AdaQN relies on the online network corresponding to the selected target network for sampling actions as a proxy for picking the best agent.
>
> We hope that this clarifies our setting as requested by the Reviewer.

---

> > ### Comment · Reviewer_Kj2W · 2024-11-28
> >
> > Thank you for the explanation.
> >
> > Equation 2 seems to describe the hyperparameter optimization setting, but does not seem to include the adaptive scenario that you are also considering in your algorithm, as the equation does not seem to take into account that the hyperparameters can be changed during optimization.
> >
> > Regarding the protocol, I have one more crucial question. Both in figure 4 and figure 7, you are considering aggregated results across multiple tasks. In these settings, for the comparison algorithms, are you evaluating each hyperparameter by considering it's average across all tasks, and trying to find the hyperparameter that performs best on average? Or are you tuning the best hyperparameters per task?

---

> > > ### Author Response · Authors · 2024-11-28
> > >
> > > Thank you for the follow-up questions.
> > >
> > > > Equation 2 [...] does not seem to include the adaptive scenario that you are also considering in your algorithm [...].
> > >
> > > We believe that by using the subscript in $\zeta_t^k$ in Equation $2$, we consider the timestep $t$ meaning that the hyperparameters can evolve during the training, thus including the adaptive scenario.
> > >
> > > > Regarding the protocol, [...] are you evaluating each hyperparameter by considering it's average across all tasks, and trying to find the hyperparameter that performs best on average? Or are you tuning the best hyperparameters per task?
> > >
> > > The answer to this question is different depending on the type of hyperparameter search spaces (finite or infinite) because this choice needs to be made according to the underlying purpose of the experiment. This is why the explanation of the evaluation protocol in Appendix $B.2$ is split into two parts.
> > >
> > > For Figure $4$, where the hyperparameter search space is finite, the goal of the experiment is to compare AdaQN against the performances of each individual hyperparameter. Therefore, we report the performance of grid search as the maximum performance of the best hyperparameter at each timestep $t$, i.e., the hyperparameter from which the performance comes can change for each timestep $t$ but not for each task. As explained in the previous answer and Appendix $B.2$, the performance of random search in this setting is not computed from a max operator, thus the distinction between per hyperparameter and per task computation is irrelevant.
> > >
> > > For Figure $7$, where the hyperparameter search space is infinite, the goal of the experiment is to see which algorithm reaches the higher performance for a given number of environment interactions. To make random search and DEHB more competitive in this setting, we keep the samples collected by the previous individual trials when a new trial starts. This means that, for each task and each seed, random search and DEHB generate a sequence of returns over $180$M environment interactions. From this sequence, we compute a running maximum over timesteps, meaning that a single peak in this sequence would lead to a plateau after applying the running maximum. Finally, we compute the IQM over seeds and tasks. We point out that it results in stronger baselines than taking the maximum per task as the maximum is taken per seed.
> > >
> > > We hope this clarifies the evaluation protocol. We now updated Appendix $B.2$ to include this clarification. We thank again the Reviewer for their thorough comments. We remain available for any follow-up questions.

---

> > > > ### Comment · Reviewer_Kj2W · 2024-11-28
> > > >
> > > > Thank you for the explanation.
> > > >
> > > > Regarding the grid search in Figure 4, I don't find the protocol of picking the best parameter on average fair. This is because the computation time that you are considering in your approximation is the computation time for trying all hyperparameters on each task. This is the same amount as needed for tuning the best hyperparameter per task, so I would expect an evaluation based on picking the best hyperparameter per task, and averaging these performances together.
> > > > I am also not convinced by the random search evaluation, because the length of random search does not necessarily have to use the same number of environment steps per hyperparameter optimization trial as AdaSAC, and it may be better to use a smaller number. I was also not convinced by your arguments about the computational cost of computing the expected maximum over a random search. Based on the learning curve data, it should be fairly easy to set up a Monte Carlo estimation that would probably converge reasonable quickly (or use some other approximation).
> > > >
> > > > On the other hand, the evaluation in Figure 7 appears fair. However, I am wondering about the setting of the trial lengths for random search and the parameters for DEHB. It seems that they can't really try many hyperparameters due to setting of the budget, so I am wondering whether they may perform better with much shorter initial trials, for example running many short optimizations for a little bit of time, then picking the best one, and optimizing it until the end. Currently, for DEHB, the shortest horizon seems to be set at 20M, and max at 40M, so it can only try a couple hyperparameters until the budget of 180M, and random search can also only try 6 hyperparameters at 30M each. I am not sure these parameter settings are optimal for the comparison algorithms. How can you be sure that your comparison methods are well tuned?

---

> > > > > ### Author Response · Authors · 2024-11-28
> > > > >
> > > > > We thank the Reviewer for the detailed suggestion.
> > > > >
> > > > > > 1.a. [...] I would expect an evaluation based on picking the best hyperparameter per task, and averaging these performances together.
> > > > >
> > > > > > 1.b. [...] Based on the learning curve data, it should be fairly easy to set up a Monte Carlo estimation that would probably converge reasonable quickly (or use some other approximation).
> > > > >
> > > > > We have decided to pick the best hyperparameter across all tasks instead of picking the best hyperparameter per task because the main goal of this experiment is to show how AdaSAC performs against fixed hyperparameters. Nevertheless, we understand the Reviewer's claim and have updated our results following the $2$ requested changes. Since uploading a revision is no longer allowed, we show Figure $4$ at https://imgur.com/a/NZawtDg. Grid search is now computed by taking the maximum over each task before computing the IQM. Additionally, we use Monte-Carlo sampling to approximate the empirical average of random search until the maximum performance has been reached. In case of acceptance, we will update Figure $4$ and Appendix $B.2$ to incorporate the requested changes.
> > > > >
> > > > > We point out that the performances of grid search slightly increased because the best hyperparameters yield similar return, as shown in Figure $5$.
> > > > >
> > > > > > 2. On the other hand, the evaluation in Figure 7 appears fair. However, [...] How can you be sure that your comparison methods are well tuned?
> > > > >
> > > > > We have observed that reducing the amount of available environment interactions can be harmful because of the Atari benchmark's complexity (visual inputs, complex dynamics, etc.). For example, during the initial phase of our experimental evaluation, we tested random search with individual trials having a budget of $10$M frames on the game of Pong, observing worse results than for $30$M frames. Thus, we have decided to keep $30$M as the value for the other games.
> > > > >
> > > > > In conclusion, limiting the first trials to a low number of environment interactions could lead to sub-optimal choices, which could lead to stagnating performance when trained with more environment interactions.

---

> ### Comment · Reviewer_Kj2W · 2024-12-01
>
> Regarding:
> >We believe that by using the subscript $t$ in Equation 2, we consider the timestep meaning that the hyperparameters can evolve during the training, thus including the adaptive scenario.
>
> I see your point, but I think it could be made clearer by including the sequence of hyperparameters into the objective. Currently, you write it in terms of optimizing at each step, but actually the objective is the best performance after the sequence of multiple hyperparameters.
>
> Regarding your new comments about the protocol:
> >because the main goal of this experiment is to show how AdaSAC performs against fixed hyperparameters.
>
> This statement doesn't seem consistent with the experiment to me. If the aim is to show performance against fixed hyperparameters, there is no need to take into account the hyperparameter search time, and you could just show the max performing hyperparameter without rescaling. The fact that you do include such rescaling indicates to me that this experiment is about the hyperparameter optimization performance. Anyhow, I appreciate that you have added the change. One remaining comment I have about this is that the performance of random search would depend on the length of the learning. It is expected that your proposed autoRL method would require more time to reach peak performance compared to just running the best hyperparameter picked by an oracle. From this, we might expect that the optimal random search trial length is shorter than the required trial length for your autoRL method. Therefore, to make the result more convincing, I think it's best if you also do a search over the trial length, and pick the trial length that maximizes the performance for random search. This would make for the most convincing results.
>
> Regarding the results in Figure 7, I was also not convinced by your arguments. It seemed to me that you played around with the settings a bit, and picked something that seemed reasonable. But, I think it would be necessary to do a principled search for the hyperoptimizer parameters and provide evidence that they are performing well. It seems that instead of replicating existing baselines (where we could check against prior performance), you tuned everything yourself, so it is unclear to me whether your baseline algorithms are well-performing. In terms of performance of the hyperoptimizers, due to your setting where you consider the performance of the current best trained policy, it is necessary to both select the best hyperparameters, and to train these hyperparameters until completion. You mentioned that short trials for random search do not work well, but this may be because the trial length is too short to reach peak performance of the policy. If you instead use short trials to search for the hyperparameter followed by training with this hyperparameter until completion, the results may be different. For the adaptive methods like DEHB, as you're only showing the learning performance, it is not clear whether the algorithm is functioning correctly. It could be just stuck evaluating various hyperparameters for 20M frames without checking the performance on any hyperparameter for 40M frame thus limiting the achieved performance. I requested also adding the performance of the best hyperparameter when training directly with this hyperparameter from scratch, yet this is not included (your method is not expected to outperform this, and it's not a problem if it does not, but it would be good to know how it compares for reference, because this result would provide an upperbound on the performance of non-adaptive hyperparameter optimization).
>
> Moreover, there are other metrics by which to check hyperoptimization performance, e.g., instead of looking at the performance of the current best policy, one could look at the performance of the policy after training to completion with the currently selected best hyperparameters. I understand that your setting is not the same, but the comparison algorithms are not necessarily designed for your setting, so I think it may be better to take into account the possibility of stopping hyperoptimization half-way through and training until the end with the chosen hyperparameter.
>
> While I am not completely convinced by the experimental work, I have decided to keep my score, and I encourage the authors to further revise the analysis.

---

> > ### Author Response · Authors · 2024-12-03
> >
> > We thank the Reviewer for the extensive feedback. With the following, we hope that we clarify the evaluation protocol. We will provide all the required explanations and details in the final version of our work.
> >
> > > 1. [...] You write it in terms of optimizing at each step, but actually the objective is the best performance after the sequence of multiple hyperparameters.
> >
> > We propose to reformulate Equation $(2)$ the following way:
> >
> > $\max_{(\zeta\_{t'})\_{t' \leq t}} J(\theta\_t) s.t. \forall t' \leq t, \theta_{t'} \in \text{arg} \min_{\theta} f(\theta; \zeta_{t'}, \theta_{t' -1})$
> >
> > where $t$ is the current timestep, $\theta_t$ is the parameters of the function approximator after $t$ timesteps, $\zeta_t$ are the hyperparameters after $t$ timesteps, $J$ is the cumulative discounted return, and $f$ represents the objective function of the learning process.
> >
> > This way, the objective is to find the sequence of hyperparameters $(\zeta_{t'})_{t' \leq t}$ that leads to the highest performances at each timestep $t$.
> >
> > > 2. [...] I think it's best if you also do a search over the trial length, and pick the trial length that maximizes the performance for random search. This would make for the most convincing results.
> >
> > As shown in Figure $4$ (right), the performance of all hyperparameters keeps increasing until the end of the training. Therefore, stopping before the proposed trial length does not lead to a stronger random search variant. In other words, the proposed trial length is already the one that maximizes the performance of random search as shown here: https://imgur.com/a/Zkwx2Xu.
> >
> > > 3. [...] I think it would be necessary to do a principled search for the hyperoptimizer parameters and provide evidence that they are performing well.
> >
> > We point out that when the hyperparameter search space is large, most hyperparameters require many environment interactions to reach high returns. Crucially, we observe empirically that in the case where the hyperparameters lead to high returns, the performances keep increasing during the trial which means that stopping earlier would not be beneficial. On the other hand, allowing too many environment interactions would prevent the algorithm from exploring the hyperparameter search space which reduces the chance of selecting "working" hyperparameters. Importantly, we observe empirically that for most seeds, at least one trial leads to high returns which means that there is no need to further reduce the trial length. This is why we argue that the chosen values for random search and DEHB's parameters are reasonable. Finally, we saw that small changes in the trial length do not drastically impact the performance.
> >
> > > 4. It seems that instead of replicating existing baselines (where we could check against prior performance), you tuned everything yourself, so it is unclear to me whether your baseline algorithms are well-performing.
> >
> > Hyperparameter optimization for DQN on the Atari benchmark is not common due to the training time required by AutoRL methods. Additionally, we chose a wide search space to further increase the difficulty of the task which made the setting unique. Given this setting, we had no other choice than to tune the algorithms ourselves. As explained in the answer to Point $3$, we chose a large trial length that is small enough so that, for most seeds, at least one trial leads to high returns.
> >
> > > 5. If you instead use short trials to search for the hyperparameter followed by training with this hyperparameter until completion, the results may be different.
> >
> > We agree that using short trial lengths first before increasing the trial length is a superior strategy compared to keeping a trial length fixed. However, this idea is not part of random search, it belongs to more sophisticated methods such as DEHB which we already include in our analysis.
> >
> > > 6. [...] [DEHB] could be just stuck evaluating various hyperparameters for 20M frames without checking the performance on any hyperparameter for 40M frame thus limiting the achieved performance.
> >
> > Thank you for pointing this out. We have verified by looking at our results that DEHB is not stuck with trials of length $20$M, and that it is using trials with $40$M frames in each game.

---

> ### Author Response · Authors · 2024-12-03
>
> > 7. I requested also adding the performance of the best hyperparameter when training directly with this hyperparameter from scratch [...].
>
> We believe that the concept of "best hyperparameter" is ill-defined in our work, because the best hyperparameters are different for each seed and each training step, due to the non-stationarity of the RL optimization procedure. Because of this, our experiments are designed to compare AutoRL methods, not fixed hyperparameters. Even when fixed hyperparameters are considered for random search and DEHB, the chosen hyperparameters vary from one seed to another. Nonetheless, the best performance obtained across the trials is observable as the final performance of random search and DEHB. This comes from the fact that we apply the maximum operator across each trial as explained in Appendix $B.2$.
>
> > 8. I understand that your setting is not the same, but the comparison algorithms are not necessarily designed for your setting, so I think it may be better to take into account the possibility of stopping hyperoptimization half-way through and training until the end with the chosen hyperparameter.
>
> We believe that SEARL suits the setting of our work well as it aims at maximizing the return by selecting well-performing hyperparameters. Moreover, we point out that performing early stopping is not straightforward as it requires many subjective choices (the number of trials before stopping, the trial length, and the selected hyperparameter for the long training) which would require large amount of computational resources to tune empirically, which goes against the aim of this work of finding effective hyperparameters while optimizing the usage of samples. This is why, we chose standard AutoRL baselines that require less tuning while offering competitive performances [1].
>
> [1] Theresa E. et. al. Hyperparameters in reinforcement learning and how to tune them. ICML, 2023.

---

### Official Review · Reviewer_FXWZ · 2024-11-06

**Soundness:** 3
**Presentation:** 3
**Contribution:** 3
**Rating:** 8
**Confidence:** 3

**Summary:**

The authors have developed a new AutoRL approach called Adaptive Q-Network (AdaQN), which aims to address the non-stationary optimization procedure in RL without requiring additional samples. The algorithm learns multiple Q-functions with different hyper-parameters while using the smallest approximation error as a shared target. So there's no need for additional samples. The paper also provides theoretical motivation that supports the algorithm and through experiments on Mujoco and Atari, AdaQN is shown to have better performance and sample efficiency.

**Strengths:**

1. Simple but novel and effective design of training multiple Q functions with different hyper-parameters settings and target selection mechanism which means no additional samples are required for hyper-parameter tuning.
2. AdaQN is shown to performance better in Mujoco and Atari benchmarks than DQN and other autoRL baselines
3. Theoretical support is also presented to support the motivation of the selection mechanism in AdaQN.
4. AdaQN can be applied to other critic-based algorithms

**Weaknesses:**

From my understanding, AdaQN's performance is still dependent on the initial range of hyper-parameters considered in the multiple Q setup. If this range is not representative of the optimal settings, AdaQN may not achieve the best possible performance. Also, it seems error or bias in the calculation of approximation error will affect the selection of the shared target.

**Questions:**

Can AdaQN effectively be applied on continuous action spaces (actor-critic frameworks) especially in complex high-dim environments? Which Q should be use to derive a policy network (maybe similar as we choose the one for target calculation)?

---

> ### Author Response · Authors · 2024-11-20
>
> We thank the Reviewer for the insightful feedback and questions.
>
> > Weakness 1: From my understanding, AdaQN's performance is still dependent on the initial range of hyper-parameters considered in the multiple Q setup. If this range is not representative of the optimal settings, AdaQN may not achieve the best possible performance.
>
> We agree that the range of available hyperparameters influences the performances. In the case that little domain knowledge on the hyperparameters is available, it is a safe choice to consider a wide range of hyperparameters. Importantly, AdaQN performs well in this case. To verify this, we evaluate AdaSAC with a wide search space in Section $5.4$. Indeed, AdaSAC could select from $18$ action functions, $24$ optimizers, $4$ losses, a wide range of learning rates, and fully parameterizable architectures (a detailed description of the search space is available in Table $4$, in Appendix $B.1$). This way, we ensure that poorly performing hyperparameters are also included in the search space. In Figure $7$ left, we conclude that AdaDQN is able to learn effectively even in wide search space, reaching higher performances than the baselines.
>
> > Weakness 2: Also, it seems error or bias in the calculation of approximation error will affect the selection of the shared target.
>
> We point out that we do not aim at estimating the approximation error directly. Instead, we use the minimum of the considered cumulated losses as an estimation of the hyperparameter minimizing the considered approximation errors. Importantly, we show in Theorem $4.1$ that this selection mechanism is accurate under the condition that the dataset is rich enough to represent the Bellman operator in expectation. To verify that this selection mechanism is effective in practice, we evaluate other variants of the proposed approach. AdaDQN-max is a variant of AdaDQN where the maximum of the cumulated loss is chosen instead of the minimum. RandDQN and RandSAC are variants of AdaDQN and RandSAC where the next target is chosen randomly from the set of online networks. The results presented in Figure $2$ (right) and Figure $4$ (right) support the idea that the proposed selection mechanism is effective, as AdaDQN and AdaSAC consistently outperform the proposed variants.
>
>
> > Question 1: Can AdaQN effectively be applied on continuous action spaces (actor-critic frameworks) especially in complex high-dim environments?
>
> Yes, we present an adaptive version of SAC (AdaSAC) in Algorithm $2$, in Appendix $B$. AdaSAC outperforms random search and grid search on $6$ MuJoCo environments. The individual performances of AdaSAC and the baselines are presented in Figure $5$. We argue that those environments can be considered complex and high-dimensional, especially *Humanoid* and *HumanoidStandup*, where the state space is composed of $348$ dimensions and the action space is composed of $17$ dimensions.
>
>
> > Question 2: Which Q should be use to derive a policy network (maybe similar as we choose the one for target calculation)?
>
> Ideally, one should choose the most accurate $Q$-network. Unfortunately, this information is not available but more importantly, learning the policy from the same $Q$-network could lead the other $Q$-networks to learn passively. This would make the accuracy of the other $Q$-networks drop as identified in [1] making them useless for the rest of the training. As suggested by the Reviewer, we take the $Q$-network corresponding to the selected target network (as a proxy for the best-performing network). To avoid passive learning, we randomly select another $Q$-network with low probability as commonly done with $\epsilon$-greedy policies. This information is explained in more detail in the last paragraph of Section $4.1$.
>
> [1] G. Ostrovski, et al. The difficulty of passive learning in deep reinforcement learning. NeurIPS, 2021.

---

### Author Response · Authors · 2024-11-20
**Additions to the revised submission**

We thank the Reviewers for their valuable feedback which helped us to improve the paper. We have submitted a revision with two additions highlighted in orange:
1. *[requested by Reviewer PnAT]* We prove the convergence of AdaQN to the optimal action-value function in the tabular setting in Appendix A.1.

2. *[requested by Reviewer Kj2W]* We explain the evaluation protocol in detail in Appendix B.2. For each method, we describe the steps leading to the reported performances in Section $5$.

---

### Meta-Review · Area_Chair_fLNn · 2024-12-21

**Metareview:**

The paper presents an autoRL technique to dynamically and automatically select the best hyperparameters in temporal different Q-network updates.  The strength of this work is an effective technique that clearly improves the stability and sample efficiency of learning.  The main weakness of this work is the computational overhead of updating an ensemble of Q-networks.  However, this computation can be easily parallelized when there is sufficient computational resources.  The approach is well motivated.  It is simple and yet effective.  It clearly advances the state of the art.

**Additional Comments On Reviewer Discussion:**

There was no additional discussion since the reviewers unanimously recommend acceptance.

---

### Decision · Program_Chairs · 2025-01-22

Accept (Poster)